# Gas Sensing Properties of Indium–Oxide–Based Field–Effect Transistor: A Review

**DOI:** 10.3390/s24186150

**Published:** 2024-09-23

**Authors:** Chengyao Liang, Zhongyu Cao, Jiongyue Hao, Shili Zhao, Yuanting Yu, Yingchun Dong, Hangyu Liu, Chun Huang, Chao Gao, Yong Zhou, Yong He

**Affiliations:** 1State Key Laboratory of Coal Mine Disaster Dynamic and Control, Chongqing University, Chongqing 400044, China; cyliang_1993@163.com; 2Key Laboratory of Optoelectronic Technology and Systems of the Education Ministry of China, College of Optoelectronic Engineering, Chongqing University, Chongqing 400044, China; haojiongyue@foxmail.com (J.H.); zsl18678631330@163.com (S.Z.); yyt_ysu@163.com (Y.Y.); 202208021016@stu.cqu.edu.cn (Y.D.); 202308021039@stu.cqu.edu.cn (H.L.); 20192547@cqu.edu.cn (C.H.); gaoc@cqu.edu.cn (C.G.); 3Department of Ultrasound, The Affiliated Hospital of Southwest Jiaotong University, The Third People′s Hospital of Chengdu, Chengdu 600031, China; czycr123@126.com

**Keywords:** field–effect transistor, gas sensor, MOSFET, TFT, In_2_O_3_

## Abstract

Excellent stability, low cost, high response, and sensitivity of indium oxide (In_2_O_3_), a metal oxide semiconductor, have been verified in the field of gas sensing. Conventional In_2_O_3_ gas sensors employ simple and easy–to–manufacture resistive components as transducers. However, the swift advancement of the Internet of Things has raised higher requirements for gas sensors based on metal oxides, primarily including lowering operating temperatures, improving selectivity, and realizing integrability. In response to these three main concerns, field–effect transistor (FET) gas sensors have garnered growing interest over the past decade. When compared with other metal oxide semiconductors, In_2_O_3_ exhibits greater carrier concentration and mobility. The property is advantageous for manufacturing FETs with exceptional electrical performance, provided that the off–state current is controlled at a sufficiently low level. This review presents the significant progress made in In_2_O_3_ FET gas sensors during the last ten years, covering typical device designs, gas sensing performance indicators, optimization techniques, and strategies for the future development based on In_2_O_3_ FET gas sensors.

## 1. Introduction

Industrialization has undergone fast iteration and modernization in recent decades, resulting in the emission of several poisonous and hazardous gases that have adversely affected air quality and placed human health at risk [1,2]. Thus, gas sensors are now utilized extensively in several domains such as medical diagnosis, food spoilage detection, agriculture, chemical production, and safety concerns related to flammability and explosive events [3,4,5,6,7,8]. Standard instruments based on gas chromatography spectroscopy and mass spectrometry can achieve extraordinary levels of detection precision [9]. However, the limitations of the conventional equipment such as high expenses, intricate procedures, and lack of portability render them inappropriate for the demands of rapid detection in some situations. Intensive study in recent years has concentrated on semiconductor resistive sensors with simple designs, considering the need for inexpensive and rapid detection. Due to their exceptional stability, accessible cost, high response, and sensitivity, metal oxide semiconductor (MOS) materials, which are utilized in mainstream commercial gas sensors, have been extensively studied as sensitive materials in semiconductor resistive gas sensors [10,11,12,13,14,15].

The rapid development of the Internet of Things (IOT) has placed more demanding requirements on gas sensors that rely on metal oxides. These requirements generally include the capability to reduce the operating temperature, enhance selectivity, and facilitate integration compatibility [16,17,18]. Over the last decade, field–effect transistor (FET) gas sensors have obtained more attention to tackle the three primary concerns stated before. The fundamental characteristic that sets field–effect transistors apart from resistive sensors is the control to channel carrier transport of gate voltages. In general, field–effect transistors can be built with reduced dimensions, making them suitable for integrating complementary metal oxide semiconductor (CMOS) circuits [19,20,21]. In addition, metal–oxide–based field–effect transistor gas sensors offer superior performance compared to resistive sensors in terms of enhanced response, increased signal–to–noise ratio, and reduced limits of detection [22,23,24,25]. The extraction of multiple parameters of FETs, including transient threshold voltage (V_th_), sub–threshold swing (SS), switching ratio (I_on_/I_off_), carrier concentration (μ), transconductance (g_m_), saturation output current, etc., can be advantageous for the identification of mixed gases. Available multi–parameters increase the data dimension by algorithms to improve higher accuracy in predicting gases’ types and concentrations, thus facilitating the creation of electronic noses for gas sensors [26,27,28]. Moreover, a back–gate top–contact configuration is commonly adopted for FET gas sensors. Since gas–sensitive materials require frequent exposure to the gas environment, the top–contact structure is less selected. The direct interaction between gas–sensitive materials and target gases facilitates expedited measurement.

Indium oxide (In_2_O_3_) is a transparent metal oxide semiconductor with a bandgap of 3.4 to 3.75 eV. It surpasses other common oxide semiconductors, such as SnO_2_, ZnO, and CuO, in carrier concentration and mobility. This high electrical conductivity results in exceptional gas sensitivity, particularly in low concentrations (ppm to ppb). Surface modifications or doping of In_2_O_3_ can enhance its selectivity for specific gases. Its stability under various environmental conditions ensures reliable and consistent performance over time. In addition to its low power consumption, In_2_O_3_ features fast response and recovery times. It is highly versatile, suitable for integration into different sensor configurations such as thin films, nanoparticles, and nanostructures. This versatility supports flexible gas sensing applications and improves FET performance by controlling the off–state current [29,30,31,32]. The optical transparency of In_2_O_3_ makes it ideal for transparent conductive layers, with indium–based oxides like IGZO being advanced materials in display technologies. Summarizing advancements in indium–based oxide gas sensors can facilitate broader deployment of In_2_O_3_ FET gas sensors. In this review, the main progress of In_2_O_3_ FET gas sensors in the past decade is introduced from several aspects. First, the development of two main types of transducers, MOSFET and thin film transistor (TFT), is separately introduced. Second, the advantages of In_2_O_3_ FET gas sensors are reflected in responsivity, sensitivity, limit of detection, signal–to–noise ratio, response and recovery times, low–temperature operation, and selectivity. Third, the optimization of FET structure, operating conditions, and algorithms is the future direction of high–performance In_2_O_3_ FET gas sensor.

## 2. Typical Transducers for In_2_O_3_ FET Gas Sensors

In_2_O_3_ resistive gas sensors have been extensively studied over the past few decades, whereas In_2_O_3_ FET gas sensors have only been gradually developed in the last decade. Research on In_2_O_3_ resistive gas sensors primarily focuses on the nanostructures of the sensing materials or their composites. Different preparation methods for In_2_O_3_ nanomaterials can lead to variations in gas sensor performance. Building on the advancements in In_2_O_3_ gas sensing materials, the development of high–performance FET gas sensors is becoming more feasible.

In 2021, Tong et al. achieved a detection limit of approximately 4 ppm for CO employing porous In_2_O_3_ nanorods fabricated via a hydrothermal method [33]. In 2022, Son et al. reported that In_2_O_3_ nanowires prepared by thermal evaporation demonstrated high sensitivity to ethanol and selective detection of common reducing gases such as acetone, CO, NH_3_, and H_2_S at room temperature [34]. Furthermore, In_2_O_3_ nanocomposites hold significant promise for improving gas sensor performance, particularly when combined with precious metals and metal oxides. In 2021, Wang et al. investigated the effects of four metal dopants (Au, Ag, Pt, and Pd) on the ethanol sensing performance of mesoporous In_2_O_3_. All four doped In_2_O_3_ variants exhibited improved responses to ethanol compared to pure In_2_O_3_, with Pd–doped In_2_O_3_ showing the highest response at the optimal operating temperature [35]. In the same year, Luo et al. enhanced hydrogen gas sensing by co–doping In_2_O_3_ nanotubes with PdO and NiO. This ternary composite achieved the highest response to hydrogen and significantly reduced response and recovery times [36]. Recent studies on In_2_O_3_–based composites for gas sensing have also highlighted emerging materials such as MXenes, carbon nanotubes, carbon nanofibers, graphene, and two–dimensional transition metal sulfides [37,38,39,40,41,42,43,44].

Some differences exist in the operational mechanisms of In_2_O_3_ resistive and FET gas sensors, although both rely on the interaction between the thin film sensing layers and target gases. For resistive gas sensors, the gas molecules react with the metal oxides, altering the resistance and thereby detecting the gas concentration. This change is measured by tracking the variation in resistance values. In contrast, for FET gas sensors, the performance changes in the gas–affected thin film sensing layer are manifested in the charge transfer of TFT devices and screening of FET and floating gate–based devices under an electric field. These changes are measured by dynamic changes in current or transient changes in voltage. Compared to the wide application scenarios of resistive sensors, FET sensors are more suitable for high sensitivity and high–selectivity gas detection. Additionally, they generally have lower power consumption and better stability. Here, we elaborate on the progress of In_2_O_3_ FET gas sensors based on two mainstream transducers (MOSFET and TFT).

### 2.1. MOSFET

Metal oxide semiconductor field–effect transistor (MOSFET) gas sensors with a floating gate structure are introduced, where the gas sensing layer, In_2_O_3_, is completely separated from the channel layer. In 2018, Seongbin Hong et al. used Pt–modified In_2_O_3_, prepared by inkjet printing, as the sensing material for a MOSFET oxygen sensor. The sensor, with optimal Pt doping concentration, significantly enhanced response below 140 °C based on physical adsorption mechanisms [45]. In the following year, their group applied Pt–modified In_2_O_3_ MOSFETs to detect CO. The sensor showed improved performance, including better responsivity and faster response and recovery times at the optimal Pt doping concentration and operating temperature [46]. Additionally, applying a positive control gate voltage (Vpre) can further enhance the responsivity to reducing gases like CO.

In 2020, Wonjun Shin et al. investigated the impact of deposition conditions on the signal–to–noise ratio of gas sensors. The MOSFET gas sensors were based on In_2_O_3_ thin films of the same thickness (30 nm) prepared by magnetron sputtering at different radio frequency powers [47]. Within the low power range (50–100 W), increased power results in larger In_2_O_3_ grains with fewer grain boundaries, consequently reducing film resistivity. However, rapid deposition induces a significant increase in defects and the generation of many small grains at excessively high powers (over 100 W), contributing to poor film uniformity and higher resistivity. Nevertheless, variations in RF sputtering power do not affect the low–frequency noise (LFN) characteristics in MOSFET gas sensors, as the channel of the MOSFET determines the low–frequency noise characteristics. The LFN characteristics of MOSFET–based gas sensors depend on factors such as the channel, operating region (linear region/sub–threshold region/saturation region), and biases between the source and drain. The optimal signal–to–noise ratio of resistive gas sensors occurs where the resistivity of the sensitive film is lowest, despite fewer defects as adsorption sites and suboptimal response. Conversely, the LFN characteristics of MOSFET gas sensors remain unaffected by the defects of the sensitive thin films and thus the maximum signal–to–noise ratio occurs at the position of maximum response since noise is determined by the channel. Therefore, the highest signal–to–noise ratio is attained at the power level where the maximum gas response occurs, attributed to the increased film defects that facilitate gas adsorption. Furthermore, their group investigated the intrinsic noise and additional noise of MOSFET gas sensors based on 12 nm In_2_O_3_ thin films in the same year. The intrinsic noise originates from the channel of the transistor, while the additional noise refers to noise generated from gas reactions. The In_2_O_3_–sensitive films prepared in varied oxygen atmospheres can lead to differences in gas sensing performance. Two types of In_2_O_3_–sensitive films prepared in Ar and Ar/O_2_ mixed atmospheres were utilized to confirm the additional noise. The MOSFET with the thin film prepared in an Ar/O_2_ mixed atmosphere demonstrated a stronger response to H_2_S without an increase in additional noise, compared to the MOSFET with the film prepared in pure Ar. Consequently, gas reactions do not introduce additional noise to the MOSFET, representing a distinct advantage over resistive sensors [48]. In the following year, their group continued the investigation on the influence of the body–source junction bias voltage on NO_2_ MOSFET sensors utilizing a 12 nm In_2_O_3_ thin film as the sensitive material. The low–frequency noise characteristics were found to be affected by the polarity of the body–source junction bias (V_BS_). Applying a forward V_BS_ of 0.5 V can reduce the 1/f noise power by approximately tenfold in the subthreshold region, resulting in a doubling of the signal–to–noise ratio (SNR) [49].

In 2021, Jung et al. fabricated MOSFET NO_2_ sensors utilizing a 10 nm In_2_O_3_ thin film as the sensitive material. This work highlighted two aspects for enhancing sensor performance. Firstly, the In_2_O_3_ MOSFET showed varying performance across different operating current ranges and demonstrated the highest responsivity in the sub–threshold region. Secondly, an increased response to NO_2_ was achieved by applying a negative pre–bias voltage, V_pre_, to the control gate [50].

In 2023, Wonjun Shin et al. investigated the issue of threshold voltage shift due to off–state stress in MOSFETs with 15 nm In_2_O_3_ thin films. The off–state stress occurs and thus the gate oxide layer is damaged by hot holes, shown in Figure 1a. When the In_2_O_3_ MOSFET is not working, with the control gate voltage below the threshold voltage, the electrical performance degradation appears, including threshold voltage negative shift and increased 1/f noise, shown in Figure 1b,c. Such instability results in variations in gas sensing performance containing responsivity and recovery time, undermining the stability and reliability of the In_2_O_3_ MOSFET gas sensors. They proposed a self–healing method based on forward biasing the body–drain PN junction, to mitigate the off–state stress–related damage. The issues stemming from the threshold voltage shift, increased noise, and reduced response to NO_2_ were induced by off–state stress–related damage. They could all be rectified using this approach, as illustrated in Figure 1d–i [51]. 

In the same year, Gyuweon Jung et al. conducted research on the controllable manipulation of adsorbed oxygen and vacancy oxygen on the surface of an 18 nm In_2_O_3_ thin film–based transistor, and the application of this manipulation in gas selectivity was demonstrated. The key to this method lies in a heater and an electric field of gate, where the types and quantities of adsorbed oxygen and vacancy oxygen on the surface of In_2_O_3_ are controlled by the polarity and magnitude of the electric field (V_GS_). Numerous electrons exist on the surface of an In_2_O_3_ thin film when V_GS_ is positive, leading to abundant oxygen adsorption, and thus In_2_O_3_ thin film primarily senses through adsorbed oxygen. Conversely, electrons flow from the surface to the gate when V_GS_ is negative, causing oxygen desorption, and the In_2_O_3_ thin film primarily senses through vacancy oxygen, as illustrated in Figure 2a. Hence, this method selectively enhances reactions with oxidizing gases (e.g., NO and NO_2_) through adsorbed oxygen and reducing gases (e.g., H_2_S, NH_3_, and CO) through vacancy oxygen, shown in Figure 2b [52]. In addition, Gyuweon Jung et al. investigated the effect of the hydroxyl group content in sensitive material, In_2_O_3_ thin film, on the NO_2_ sensing performance of MOSFETs. A 30 nm–thick In_2_O_3_ thin film was prepared via radio frequency magnetron sputtering, with the hydroxyl groups being enriched by introducing oxygen during the sputtering process (Ar:O_2_ = 30:1). The hydroxyl–enriched In_2_O_3_ thin film (A1) exhibited a higher work function of 4.92 eV and a larger Debye length over the pristine In_2_O_3_ thin film (A0), and demonstrated superior NO_2_ sensing performance, whether utilizing MOSFETs or resistive sensors as transducers. Specifically, the In_2_O_3_ MOSFET with the enriched hydroxyl groups even achieved higher response and lower detection concentration for NO_2_ at a lower temperature (100 °C), as shown in Figure 2h [53].

The performance of In_2_O_3_ MOSFETs gas sensors is also improved by optimization of the device structure. In 2023, Seongbin Hong et al. prepared 12 nm In_2_O_3_ thin film–based MOSFETs with different channel aspect ratios and channel areas to optimize the sensing performance of NO_2_ and H_2_S. The macroscopic structural parameters of the sensors significantly affected the electrical performance of the transistors, further influencing the gas sensing performance. Firstly, a larger aspect ratio resulted in greater saturation output current, larger transconductance, as shown in Figure 3a,b, and increased response to H_2_S and NO_2_. Secondly, a smaller channel area led to increased response to H_2_S and NO_2_ when keeping the aspect ratio consistent, as illustrated in Figure 3c,d [54]. Jinwoo Park et al. replaced the commonly adopted metal gate in MOSFET gas sensors constructed with a 15 nm–thick In_2_O_3_ film with a polycrystalline silicon gate. The improved gates were designed as two ends, V_CG1_ and V_CG2_. The In_2_O_3_ MOSFET exhibited better electrical performance in terms of a larger output current and transconductance when V_CG2_ is open, in comparison with the case where V_CG2_ is grounded, as shown in Figure 3e,f. Absolutely, the device demonstrated a greater response to H_2_S when V_CG2_ is open, as shown in Figure 3g [55].

### 2.2. TFT

Unlike the MOSFET with a floating gate structure mentioned above, In_2_O_3_ thin films play a dual role as both the sensing layer for the gas reaction and the channel layer for carrier transport for thin film transistor (TFT)–type gas sensors, which generally adopt a back–gate structure. In 2012, M. Seetha and D. Mangalaraj prepared ethanol TFT sensors based on porous indium oxide films using indium chloride solution as a precursor, followed by an annealing process. Although the porous film structure resulted in suboptimal TFT electrical performance, the TFT based on this porous In_2_O_3_ film achieved selective detection of ethanol at room temperature [56]. The influence of different annealing times at 400 °C on the electrical performance of TFTs based on spin–coated oil–soluble In_2_O_3_ nanoparticles was reported by Wang et al. in 2015. The In_2_O_3_ TFT annealed for 10 min exhibited the highest output current in air [57]. In 2017, Shariati Mohsen demonstrated that the morphology of Sn–doped In_2_O_3_ nanowires was affected by annealing temperature, which in turn influenced the electrical performance of the TFT as well as the recovery process following H_2_S exposure. A laser excitation method was employed to achieve rapid response and recovery within seconds for 20 ppb H_2_S [58]. In 2020, Jun et al. fabricated ethanol sensors based on InYbO nanofiber FETs. Similarly, XPS analysis of the O 1s spectrum was conducted to confirm the oxygen vacancies’ sites for gas reactions and the electrical performance including switching ratio, carrier concentration, sub–threshold swing, and threshold voltage, which were explored based on In_2_O_3_ nanofibers TFT with different Yb doping concentrations. The InYbO TFT with the optimal Yb doping ratio achieved detection of 1 ppm ethanol at low temperatures [59]. In the same year, Chen et al. fabricated TFT gas sensors based on Yb–doped In_2_O_3_ (InYbO) nanofibers by electrospinning again. The doping of Yb increased the specific surface area of the nanofibers, resulting in more adsorption sites. Therefore, the InYbO TFT realized a significant response to a DMF as low as 89 ppb [60]. Subsequently, in 2022, Li et al. investigated the influence of Nd doping on the specific surface area and oxygen vacancies of In_2_O_3_ nanofibers. Moderate Nd doping significantly increased the specific surface area, which is beneficial for gas sensing. However, Nd doping reduced the quantity of oxygen vacancies. An optimal Nd doping ratio was determined taking these two factors mentioned above into consideration. A TFT based on 3% InNdO exhibited extraordinary sensing performance for acetone, including large responsivity, response and recovery times within tens of seconds, and excellent selectivity in organic gases [61]. Following this work, Seung Gi Seo et al. reported h–In_2_O_3_/SWNT heterojunction TFT sensors towards NO_2_ on Si or flexible substrate, which can improve the poor recovery of SWNTs at room temperature. The 2D electron gas induced by the heterojunction and effective electron compensation mechanism contribute to the recoverability [62]. In 2023, Gyuho Yeom et al. proposed a method to reduce the operating temperature of both the response and recovery processes. The approach of applying an “erase voltage” to the control gate before the reaction process is employed to enhance the response of In_2_O_3_ TFTs to NO_2_. The enhanced ERS method involves biasing to transfer the stored electrons on the dynamic gate electrode to the thin film, enabling numerous electrons to rapidly react with NO_2_. However, this method leads to a decrease in both the speed and extent of recovery. Furthermore, the pulse bias recovery (PBR) method was introduced to address this issue. Specifically, a negative bias to the control gate was applied, causing electrons to flow back to the floating gate, thereby promoting rapid desorption of NO_2_^−^ [63].

Those sensors based on FET channels typically adopt a traditional structure consisting mainly of a source, drain, and gate, and the conductivity of the FET channel is controlled by the gate voltage. In contrast, an additional insulated floating gate is added on top of the traditional structure for FET devices based on floating gate, allowing charges to accumulate or release on the floating gate, which directly affects the current conduction. Therefore, the carrier transport in the channel is regulated by both control voltage, V_CG_, and floating voltage, V_FG_. Correspondingly, floating gate FET gas sensors are more suitable for high–sensitivity gas detection. On the other hand, TFTs as the most common representative utilized in the fielding of gas sensing acquire a simpler structure, while floating gate FET sensors typically employ MOSFETs, which involve more complex manufacturing processes. MOSFETs are regularly smaller than TFTs, making them more suitable for integration. However, TFTs exhibit larger responses and higher signal–to–noise ratios, resulting in lower detection limits for gas sensing. Consequently, the specific selection between MOSFETs and TFTs is determined by considering the requirements of the actual application.

## 3. Properties of In_2_O_3_ FET Gas Sensors

In_2_O_3_–based resistive gas sensors have been extensively studied, and the evaluation of gas sensing performance typically includes indicators such as responsivity, sensitivity, noise, response and recovery times, operating temperature range, selectivity, reproducibility, linearity and power consumption, etc. [64,65,66,67,68,69,70,71,72,73]. In_2_O_3_–based FET gas sensors keep advantages in the following five aspects due to the high carrier concentration and mobility, and research during the past decade validated their wide applications. (1) Responsivity, sensitivity, limit of detection: a larger response from a gas sensor correlates with higher sensitivity and a lower limit of detection. Generally, larger responsivity means there are more easily detectable signals which are thus regarded as an essential indicator for gas sensors. (2) Signal–to–noise ratio: defined as the ratio of the intensity of the received useful signal to the background noise. Increased responsivity, typically arising from abundant adsorption sites, is frequently observed in research; however, these sites may also indicate a greater presence of defects in the FET channel, potentially resulting in increased noise and degradation of electrical performance. Therefore, responsivity cannot fully evaluate the performance of the gas sensor, and supplementary consideration should be given to signal–to–noise ratio to design gas sensors with optimal performance. (3) Fast response of recovery time: the response/recovery times are mostly determined by the gas adsorption energy, and they are typically opposite. A rapid response generally represents a slow response, and vice versa. Optimal response or recovery time can be achieved through modification of sensitive materials, adjustment of device structure, and external excitation of the device. (4) Low–temperature detection capability: metal oxide semiconductors ordinarily work at high temperatures, and reducing the operating temperature is one of the significant directions of extensive research. (5) Selectivity: this remains one of the most challenging issues for metal oxide semiconductor gas sensors. Adequate attention has been focused on doping, modification, and other material engineering methods on sensitive materials in resistive gas sensors, to settle this issue. In the field of FET gas sensors, more approaches can be attempted such as improved device structure, optimized operating conditions, and even algorithms.

### 3.1. Responsivity, Sensitivity, and Limit of Detection

The routine definition of the response for FET gas sensors is the ratio of drain current before and after exposure to the test gas. Correspondingly, changes in mobility, transconductance, switching ratio, sub–threshold swing, threshold voltage, etc. can also be labeled as responsivity. Sensitivity refers to the change in response value caused by the change in a unit concentration of the test gas. The limit of detection is divided into experimental and theoretical values, and the theoretical value is calculated according to sensitivity and signal–to–noise ratio.

In 2020, Jun et al. prepared ethanol FET gas sensors based on Yb–doped In_2_O_3_ (InYbO) nanofiber by electrospinning, with the channel consisting of a network of one–dimensional nanofibers [59]. The optimal doping ratio of 4% Yb improved the electrical properties including the positive bias stress (PBS) stability, high mobility of 6.67 cm^2^ V^−1^ s^−1^, threshold voltage of 3.27 V, and appropriate switching ratio of 10^7^. The enhanced PBS stability made the dynamic measurement of gas sensing performance more reliable. The amplification of the responsivity towards the ethanol of the Yb–doped In_2_O_3_ FET was remarkable over the resistive sensor based on the same sensitive material, as shown in Figure 4. The limit of detection of resistive sensors only reached 10 ppm, while that of TFT could reach 1 ppm. The significant improvement in sensitivity, especially for the detection capability at a low concentration, was demonstrated.

In the following year, Jung et al. utilized RF magnetron sputtering (150 W, Ar:O_2_ ratio of 10:1) to prepare a 10 nm–thick In_2_O_3_ film as a NO_2_–sensitive material. Resistive and MOSFET sensors are employed as transducers, and their response differences based on different operational conditions are studied. The response of the resistive gas sensor was insensitive to the variable operating current range and remained consistent, while the In_2_O_3_ FET exhibited the maximum response in the sub–threshold region. The current response value of the In_2_O_3_ FET could reach 3.8 × 10^4^, which is 8.15 times that of the corresponding resistive sensor. Furthermore, the response to NO_2_ of the In_2_O_3_ FET can be further increased, by applying a negative pre–bias voltage V_pre_ to the gate before the measurement [50].

In 2022, Seung Gi Seo et al. reported indium oxide and single–walled carbon nanotube heterostructures (h–In_2_O_3_/SWNT) synthesized by a one–pot hydrothermal approach, and demonstrated the fabrication of the h–In_2_O_3_/SWNT thin film transistor (TFT) NO_2_ gas sensors through drop–casting on silicon and flexible substrates. The amplification of response values of the h–In_2_O_3_/SWNT TFT was noticeable compared to the SWNT–based TFT, as illustrated in Figure 5. The h–In_2_O_3_/SWNT TFT achieved a limit of detection of 1 ppm NO_2_ and exhibited negligible degradation during exposure at nine consecutive cycles within 2 days. Moreover, the electrical reliability and reproducibility of the h–In_2_O_3_/SWNT TFT on flexible substrates were thoroughly validated even under repeated tensile strains of 0.88% [62].

### 3.2. Signal–to–Noise Ratio

The signal–to–noise ratio (SNR) refers to the ratio of the intensity of the received useful signal to that of the interfering noise. In the context of FET gas sensors, the discussion of SNR typically revolves around the low–frequency noise characteristics which may impact the accuracy and stability of the sensor. Therefore, the performance of the FET gas sensors can be enhanced by understanding and controlling the low–frequency noise characteristics.

In 2020, Wonjun Shin et al. investigated the effects of radio frequency sputtering power on the deposition rate and resistivity of the formed In_2_O_3_ thin films. The resistive sensor and MOSFET are adopted as transducers to detect H_2_S, aiming at exploring the influence on signal–to–noise ratio (SNR). The highest SNR observed in the In_2_O_3_ thin film–based resistive sensor occurred at the sputtering power that yielded the lowest resistivity, despite the device’s response not being maximal under these conditions. However, the deposition rate and resistivity, affected by the sputtering power, did not impact the low–frequency noise characteristics of the In_2_O_3_ MOSFET, which is determined by the channel of the MOSFET. Accordingly, the SNR of the MOSFET based on radio frequency sputtered.

In_2_O_3_ thin films outperformed that of a resistive–type device, as shown in Figure 6 [47]. In the same year, their group prepared 12nm–thick In_2_O_3_ thin films through RF radio frequency magnetron sputtering (5 mTorr, Ar:O_2_ ratio of 10:1) as H_2_S–sensitive material. Both the resistive sensor and MOSFET were employed to further study the low–frequency noise characteristics, mainly focusing on differentiating between intrinsic noise and additional noise caused by gas reactions. Similar to the previous work, the noise characteristics of resistive gas sensors were dominated by the polycrystalline sensing layer and influenced by the deposition conditions of the sensing material, whereas the intrinsic noise of the In_2_O_3_ MOSFET depended on the channel, and was thus not affected by the sensing material. The intensity of low–frequency noise in the sub–threshold region of the MOSFET based on RF–sputtered In_2_O_3_ thin films is approximately ten times lower than that of the resistive sensor. Furthermore, additional noise, generated from gas reactions, was measured through In_2_O_3_ thin films prepared with different Ar to O_2_ ratios. The thin films formed in atmospheres with higher O_2_ ratios exhibited stronger reactivity towards H_2_S. The gas–to–air noise ratio (GANR) values for resistive sensors reacting with H_2_S prepared in Ar and Ar/O_2_ mixed atmospheres were 1.9–2.3 and 3.8–4.1, respectively, indicating a significant increase in additional noise owing to gas reactions. In contrast, the corresponding GANR values for MOSFETs were both approximately 1, arguing that almost no additional noise increased. In summary, the MOSFET based on RF–sputtered In_2_O_3_ thin films demonstrated lower intrinsic noise and even no additional noise than corresponding resistive gas sensors, as shown in Figure 7 [48].

In the following year, their group fabricated 12 nm–thick In_2_O_3_ thin films by RF radio frequency magnetron sputtering (50 W, 5 mTorr) again as NO_2_–sensitive materials, with MOSFET as the transducer, studying the effect of bulk–source junction bias voltage on signal–to–noise ratio (SNR). Carrier number fluctuations dominated the 1/f noise of the FET device in all operational regions and the 1/f noise decreased by a factor of about 10 when an appropriate V_BS_ = 0.5 V was applied [49]. Meanwhile, the SNR doubled, implying that the limit of detection of NO_2_ was reduced by half, from 0.55 ppb to 0.27 ppb, as illustrated in Figure 8. The research conducted by the group to enhance the signal–to–noise ratio evolved from material deposition to transducer comparison, ultimately culminating in the optimization of the transducer’s operating conditions.

### 3.3. Response and Recovery

Response and recovery times are significant indicators for gas sensors and there is a high demand for fast response and recovery in various scenarios such as toxic, hazardous, and explosive prevention. In 2019, Seongbin Hong et al. applied Pt–modified In_2_O_3_ to MOSFET utilizing inkjet printing technology. The optimal 5% Pt doping concentration and the best sensing temperature of 200 °C for detecting CO were determined. The response, recovery time, and responsivity of the Pt–In_2_O were significantly enhanced. Furthermore, the response time can be further reduced by applying V_pre_ to the control gate, albeit at the expense of a certain degree of responsivity. The recovery time can be shortened to approximately 10 s when V_pre_ was fixed at −2 V, as shown in Figure 9 [46].

In 2022, Li et al. fabricated an acetone gas sensor based on Nd–doped In_2_O_3_ (InNdO) FET and the channel was composed of 1D InNdO nanofibers with a network structure assumed by electrospinning. The optimal doping ratio of Nd was determined to be 3%, considering the specific surface area of synthetic nanofibers, switching the ratio and carrier concentration of the InNdO TFT. The sensor exhibited outstanding electrical performance, characterized by a high mobility of 5.5 cm^2^ V^−1^ s^−1^ and an on/off current ratio of 10^7^. In addition, the 3% InNdO TFT exhibited a high response of 88 to 4 ppm acetone at room temperature, with a response time of 31 s and a recovery time of 53 s, as illustrated in Figure 10 [61].

In 2023, Gyuho Yeom et al. prepared 12 nm In_2_O_3_ thin films by RF magnetron sputtering (5 mTorr, 50 W, Ar:O_2_ ratio of 15:1) to detect NO_2_. A TFT with a floating gate as the transducer was utilized to detect NO_2_ with rapid response and recovery at 75 °C. The swift response was attributed to the erase bias (V_ers_) method, which transferred electrons stored on the floating gate to the sensitive film, enabling sufficient electrons to react quickly with NO_2_, and enhancing the responsivity of the sensor simultaneously. Moreover, the fast recovery was facilitated by the pulse bias method. Electrons transferred back to the gate when applying a negative bias to the control gate, leading to rapid desorption of NO_2_^−^ and NO_2_ could desorb quickly even at a low temperature of 75 °C, as shown in Figure 11. Obviously, the greater the pulse bias, the faster the desorption rate of NO_2_ [63]. Regarding the realization of fast response and recovery times in gas sensors, more attention is focused on material modification in addition to traditional external excitation methods such as lighting and heating. Furthermore, FET devices can also be optimized by adjusting operational conditions.

### 3.4. Low–Temperature Detection Capability

Metal oxide gas sensors typically require operation at high temperatures, which is not conducive to energy conservation and environmental protection, especially in long–term applications such as gas monitoring systems. Long–term high–temperature operation can even pose extra risks in some cases. Conversely, reducing the operating temperature can solve these issues and expand the range of applications. Therefore, achieving low–temperature detection capability for metal oxide gas sensors is one of the core pursuits in this field.

In 2012, porous indium oxide–based TFT sensors achieved acceptable ethanol selectivity at temperatures below 100 °C. In 2013, gold nanoparticle–modified Mg–In_2_O_3_ nanowire FET arrays realized the recognition of CO in mixed gas at room temperature [56,74]. In 2015, Wang et al. dissolved oily In_2_O_3_ nanoparticles in organic solvents and prepared the In_2_O_3_ TFT by spin–coating and annealing on silicon substrates. Significant changes were observed in electron mobility and current switching ratio by comparing the electrical characteristics of the TFT in air and nitrogen, indicating the sensitivity of the In_2_O_3_ TFT to O_2_ at room temperature [57]. In 2017, a TFT with a 1D nanowire mesh structure based on Sn–doped In_2_O_3_ as the conductive channel achieved the detection of 20 ppm H_2_S with a dynamic response time of several seconds at room temperature, which was assisted by laser [58]. In 2018, Seongbin Hong et al. prepared Pt–doped In_2_O_3_ MOSFET O_2_ sensors utilizing inkjet printing. Oxygen sensing at room temperature was achieved after optimizing the Pt doping concentration. The room–temperature oxygen–sensing behavior was due to physical adsorption, thus exhibiting excellent repeatability, as shown in Figure 12. Additionally, pulsed bias voltages that typically affect chemical adsorption had no effect on physical adsorption [45]. In 2020, an ethanol gas sensor based on Yb–doped In_2_O_3_ (InYbO) nanofiber TFT achieved a detection limit of 1 ppm for ethanol at temperatures below 80 °C. The 3% InNdO TFT managed a high response of 88 to 4 ppm acetone at room temperature, with a response time of 31 s and a recovery time of 53 s [59]. In 2022, the TFT gas sensor based on Yb–doped In_2_O_3_ (InYbO) nanofibers for detecting DMF also demonstrated rapid response and recovery to 89–2000 ppb DMF below 80 °C [60]. In the same year, a h–In_2_O_3_/SWNT FET gas sensor based on silicon and flexible substrate realized a detection limit of 1 ppm NO_2_ at room temperature [62]. In 2023, a NO_2_ gas sensor demonstrating rapid response and recovery capabilities was developed at 75 °C through the electron transfer mechanism between the gate and channel, attributed to the rapid desorption of NO_2_^−^ resulting from electron motion under the influence of an electric field [63]. The advancements in gas sensing behavior at room temperature mainly stem from material modifications, including preparation methods and composites. Traditional external stimuli such as light exposure have been less explored in the field of FET gas sensors in recent years, while the ability to achieve this goal through gate voltage modulation of FET devices is gradually gaining momentum.

### 3.5. Selectivity

The selectivity of a gas sensor refers to its ability to identify specific gases and it depends on the working principles, materials, and design. The following are common methods to enhance the selectivity of gas sensors. First, the selection of sensitive materials is universal. Different gas sensing elements exhibit varying selectivity towards different gases and therefore choosing the appropriate sensing material can enhance the ability to recognize the target gas. Second, filtering techniques by adsorbents or membranes are beneficial for eliminating interfering gases. A selective barrier can be established around the sensor, allowing only the target gas to pass through, thereby enhancing selectivity. Third, changing the sensing temperature not only alters the sensing mode of physical or chemical adsorption on the sensor, but also modifies the mechanisms involved in chemical reactions. Hence, the selectivity can be controlled by the operating temperature. Fourth, the data processing algorithm meets the requirements of electronic nose. The accuracy and selectivity can be improved by utilizing data processing algorithms to analyze the fluctuating signals collected by the sensor. Fifth, gas sensor arrays and differential signal processing can be employed to eliminate interference and enhance selectivity. Sixth, gate voltage can provide additional modulation of internal charge carriers and enable selective design arrangements for FET gas sensors.

In 2012, M. Seetha and D. Mangalaraj prepared an ethanol TFT sensor based on porous indium oxide thin films using indium chloride solution as a precursor through a drop–coating method. The sensor demonstrated a significantly higher response to ethanol compared to ammonia and acetone, exhibiting minimal response variation within the relative humidity range of 40–60% [56]. In 2013, Zou et al. proposed an FET gas sensor with specific selectivity for different gases based on a quasi–one–dimensional Mg–doped indium oxide nanostructure modified with different metal nanoparticles. The traditional In_2_O_3_ FET gas sensor reacts to both oxidizing and reducing gases, but the reported E–mode of the prepared FET device is unable to respond to oxidizing gases due to the high threshold voltage, as shown in Figure 13a. Therefore, one–to–one selectivity enhancement for reducing gases can be achieved by intentionally modifying the channel–sensitive materials with special metal nanoparticles. The Mg–In_2_O_3_ nanowire FET modified with gold nanoparticles exhibited a response of over three orders of magnitude to 100 ppm CO exposed in a mixed gas under E–mode, with a response time of 4 s and a detection limit of 500 ppb. Similarly, silver nanoparticles and platinum nanoparticles–modified Mg–In_2_O_3_ nanowire FETs showed enhanced selectivity towards ethanol and hydrogen under E–mode, as shown in Figure 13b,c [74]. In 2020, Chen et al. prepared a TFT gas sensor towards N, N–Dimethylformamide (DMF) utilizing the electrospinning method, with the channel material composed of a network structure of one–dimensional nanofibers. The annealed nanofibers possessed a high specific surface area, and the FET devices based on nanofibers exhibited excellent electrical performance. Hence, the sensor displayed significant response to 89–2000 ppb DMF with response and recovery times of 36 and 67 s, respectively. Specifically, exceptional selectivity towards DMF was proved among the testing of five organic gases containing DMF, toluene, xylene, ethanol, and acetone, as illustrated in Figure 13 (right graph side) [60]. FET gas sensors can achieve enhanced selectivity not only through material considerations but also through a synergistic combination of mechanism–based and material–based designs, contrasting with the extensively researched resistive sensors.

## 4. Performance Optimization of In_2_O_3_ FET Gas Sensors

### 4.1. Optimization of FET Struture

The application of In_2_O_3_–based gas sensors is well established, but most current research predominantly concentrates on the modification of sensitive materials. The structure of gas sensors based on In_2_O_3_ FETs, including gate materials, source–drain contacts, built–in heaters, and electric fields, significantly influences the performance of gas sensing. It has been mentioned earlier that polycrystalline silicon gates can enhance the output saturation current, transconductance, and response to H_2_S compared to conventional metal gates in MOSFETs [55]. In 2017, Zeng et al. introduced heavily doped ATO (antimony–doped tin oxide) as drain and source electrodes in contact with metal oxide nanowires, demonstrating more stable and lower contact resistance, in comparison with the situations based on traditional Ti contacts, as shown in Figure 14e. Moreover, the stability and low contact resistance of ATO persisted for at least 1960 h in open–air environments at 200 °C while sustaining sensitivity and response to NO_2_ without degradation. The optimization in long–term stability has been realized by replacing the source–drain conventional metal materials with metal oxides of the FET gas sensor, as illustrated in Figure 14f–h [75]. Additionally, the MOSFETs featuring a floating gate structure demonstrate the potential applications of FET gas sensors [45,46,47,48,49,50,51,52,53,54,55]. Furthermore, even parameters such as the aspect ratio and width of the cross–finger electrodes in FET channels have an impact on device performance and gas sensing properties [54].

The optimization of the channel for TFT gas sensors is crucial, since it serves as both the carrier transport layer and the target gas–sensitive layer. Therefore, the channel not only determines the electrical performance of FETs but also fundamentally influences the reactions with target gases. One–dimensional (1D) nanowires offer advantages as the channels of FET gas sensors for several reasons. First, the high surface area–to–volume ratio of 1D nanowires can increase the interaction sites with gas molecules, leading to improved sensitivity in gas sensing applications. Second, charge carrier mobility can be improved due to their small size and high crystallinity and 1D nanowires can facilitate efficient charge carrier transport, resulting in enhanced electrical conductivity and device performance. Third, properties such as conductivity and bandgap are tunable and can be tailored by adjusting size, composition, and morphology, enabling optimization for specific application requirements. Fourth, 1D nanowires can be integrated into scalable fabrication processes compatible with existing semiconductor technologies, enabling their incorporation into complex electronic devices and systems. Overall, 1D nanowires present promising opportunities for realizing high–performance FET channels with enhanced functionality and versatility.

In 2013, Zou et al. proposed a “one–key to one–lock” hybrid sensor configuration based on 1D Mg–In_2_O_3_ nanowires, for selective detection of specific reducing gases in complex environmental backgrounds. Traditional In_2_O_3_ FETs in D–mode respond to both oxidizing and reducing gases. However, the threshold voltage of the FET can be increased to form an E–mode by adjusting the Mg doping concentration, in which the device does not respond to oxidizing gases. Further selective detection of reducing gases is achieved by specific modifications with different metal nanoparticles. Specifically, the nanowires modified with Au, Ag, and Pt nanoparticles were sensitive to CO, ethanol, and H_2_, correspondingly [74].

One–dimensional metal oxide nanowires serve as excellent functional units for integrated and transparent electronic devices. However, the synthesis methods are typically uncontrollable and involve random distribution, which pose challenges for mass production. In 2019, Tae–Sik Kim et al. introduced a large–scale direct printing approach for versatile nanoscale metal oxide nanowire electronic devices. This approach offers several advantages such as highly aligned, digitally controlled, and arbitrarily long metal oxide wire (MOW) arrays, as shown in Figure 15. In addition, the applications in gas sensing were verified. The method involves high–pressure direct printing of sacrificial polymer and precursor solutions onto a silicon substrate, followed by sintering to form the MOW array. The top contact electrode is fabricated using the conventional metal thermal evaporation process. Large–scale manufacturing and precise control of the MOW process can be achieved through this direct printing approach, leading to improved stability and reliability of FET gas sensors [76]. Large–area controllable preparation of 1D nanowires is one of the future development directions.

### 4.2. Optimization of Operating Conditions

Optimizing the operating conditions of sensors to enhance gas sensing performance is a common approach for FETs. The design of FETs aims to amplify the current under the influence of the gate electrical field, which effectively amplifies the gas sensing response. Firstly, the positive or negative bias of the control gate voltage, V_pre_, applied before gas measurement, can selectively enhance the response to oxidative gases such as NO_2_ or reductive gases such as H_2_S. Additionally, selecting the appropriate polarity of gate voltage bias after gas testing can accelerate the desorption and reduce recovery time. Secondly, integrating a microheater into FETs facilitates the operation at optimal temperature to achieve the best response, sensitivity, and response or recovery time. Thirdly, biasing through pulse voltage instead of DC voltage can generally improve the electrical performance of FETs, thereby enhancing gas sensing performance. Fourthly, appropriate biasing of the body–source junction can not only increase the signal–to–noise ratio of the MOSFET gas sensors, but also partially repair repeatability and stability caused by off–state stress damage. Finally, external stimuli such as light exposure can greatly enhance gas sensing performance.

### 4.3. Algorithm

In 2021, Dongseok Kwon et al. utilized the same sensor to predict the concentrations of NO_2_ and H_2_S by training algorithms involving Recurrent Neural Networks (RNNs) and Fully Connected Neural Networks (FCNNs), with the transient responses reflecting the concentrations of target gases and operating temperatures. Even with fewer input neurons, RNN with an error rate of 1.94% outperformed FCNN with an error rate of 3.02% following the training of RNN and FCNN in the PyTorch framework [77]. Subsequent to this work, their group fabricated a MOSFET based on In_2_O_3_ thin film, and the responses towards NO_2_ and H_2_S under different operating temperatures, gate voltages, and gas concentrations were utilized as the training set. A low prediction error rate of approximately 3% for gas concentration was achieved through the employment of the Spiking Neural Networks (SNN) algorithm. Furthermore, the operational time of the SNN was around 5 s, making it suitable for rapid detection [78]. Both endeavors concerning gas concentration prediction, as shown in Figure 16, were based on the dynamic transient response of FET gas sensors, akin to the parameter extraction of resistive sensors.

Extraction of multiple parameters from transfer and output characteristics of FET gas sensors offers advantages in predicting gas types and concentrations through algorithms. Unlike resistive gas sensors relying on transient dynamic response parameters such as response values and response times, FET gas sensors offer a wider range of extractable parameters from the transient transfer and output curves such as saturation output current, switching ratio, threshold voltage, carrier concentration and mobility, sub–threshold swing, etc. Variations in gate bias lead to changes in multiple measurement parameters, and these parameters can provide independent or partially independent outputs, resulting in two–dimensional and three–dimensional response distributions. Predictions based on three–dimensional distributions generally yield more accurate gas concentration estimates compared to those based on two–dimensional distributions [79,80].

## 5. Challenges and Future Perspectives

The high carrier concentration and mobility of indium oxide are key reasons for its high gas sensing sensitivity. Table 1 lists several types of In_2_O_3_ transistors with high mobility. In_2_O_3_ FET gas sensors have garnered significant interest owing to the potential for high sensitivity, low power consumption, and compatibility with integrated circuit technology [59,62,76]. However, they also face challenges like any technology and offer future perspectives.

A principal challenge lies in achieving high selectivity for specific target gases while effectively minimizing interference from other environmental gases. Enhancing selectivity often requires the integration of additional sensing materials or advanced signal processing techniques [59,60,61]. Firstly, various preparation methods yield diverse material and gas sensing characteristics for materials primarily composed of In_2_O_3_. Incorporation of metallic nanoparticles, rare earth element doping, common metal oxide composites, and composites including carbon–based compounds, graphene, and MXene, etc. enhance the selectivity to some extent. Therefore, continued research in material engineering including exploring novel nanostructures, doping techniques, and surface functionalization methods is crucial for enhancing the selectivity of In_2_O_3_ FET gas sensors [46,50]. Secondly, external stimuli such as temperature and illumination can alter the sensing modes of gas reactions through physical or chemical adsorption. Consequently, selectivity can be controlled in specific scenarios by this method [45]. Thirdly, the utilization of gate voltage can provide additional modulation of charge carriers in the channel, thereby enhancing or suppressing the reactions towards oxidative or reductive gases [46,50,52]. Fourthly, the principal component analysis (PCA) implemented through algorithms on the test data of gas sensors is a mainstream method to effectively improve the selectivity of In_2_O_3_ FET. The multi–parameter extraction of FETs is beneficial for achieving the three–dimensional distribution of PCA and more accurately predicting gas types and concentrations [77,78,79,80].

Although the stable properties of In_2_O_3_ have been verified, the long–term stability of the composites needs to be investigated. These FET gas sensors may experience performance degradation over time due to environmental factors such as humidity, temperature variations, and exposure to target gases. Ensuring long–term stability and reliability is crucial for practical applications [62]. Another unstable factor is the contact between the source/drain electrodes and the channel material, In_2_O_3_. Ti or Cr is commonly adopted as a direct contact metal with In_2_O_3_ due to the influence of contact resistance. However, these two metals can be oxidized over a period, leading to instability and unreliability of FETs. Studies on doped metal oxides replacing the conventional Ti/Cr demonstrated more stable source/drain contact. Therefore, further exploration to investigate the stable contact is necessary [75].

Despite advancements in miniaturizing In_2_O_3_ FET gas sensors, additional efforts are required to integrate them into compact and portable devices for real–world applications. This requires addressing challenges related to fabrication techniques, signal processing circuitry, and power management. Generally, MOSFETs are smaller than TFTs, making them more conducive to integration. TFTs exhibit larger responses and higher signal–to–noise ratios, and smaller limits of gas detection. The specific type of sensor is determined by comprehensively considering the requirements in practical applications [53,76].

Enhancing both response and recovery times is crucial for applications that demand rapid detection and monitoring of gas concentrations. The rate and amount of transferred electrons, which influence the response and recovery times, plays a critical role in the oxidation or reduction reactions frequently occurring in gas sensors. Optimization of response and recovery times can be achieved through material engineering, such as doping, modification, composites, external stimuli, and internal electric fields via gate voltage [46,61,63].

In_2_O_3_ FET gas sensors regularly exhibit temperature–dependent behavior, which significantly impacts their performance and reliability, especially in fluctuating or extreme temperature environments. Developing strategies to mitigate temperature effects or implementing temperature compensation techniques is essential. The prolonged operation at high temperatures adversely affects the stability, reliability, and lifespan of the device. Therefore, achieving the target gas sensing at low temperatures or even at room temperature becomes particularly crucial. Advances in material engineering, combined with the internal electric fields generated by field–effect properties, present opportunities to achieve sensing at lower temperatures [56,74,85,86].

Overall, future perspectives of In_2_O_3_ FET gas sensors should emphasize material engineering, intelligent sensing systems, flexible and wearable devices, and interdisciplinary collaboration. Ongoing research in material engineering is crucial for enhancing the sensing performance of In_2_O_3_ FET gas sensors, which includes exploring novel nanostructures, doping techniques, and surface functionalization methods to improve sensitivity, selectivity, and stability. Integration with advanced signal processing algorithms, machine learning techniques, and wireless communication capabilities can enable the development of smart sensing systems capable of real–time monitoring, data analytics and autonomous decision–making. Expanding the application scope of In_2_O_3_ FET gas sensors to flexible and wearable devices has the potential to revolutionize healthcare, environmental monitoring, and personal safety applications. Research efforts should focus on developing flexible substrates, scalable fabrication processes, and biocompatible sensor interfaces. Collaborations among researchers in materials science, electrical engineering, chemistry, and other related fields are essential for driving innovation in In_2_O_3_ FET gas sensor technology. Interdisciplinary approaches can lead to breakthroughs in sensor design, fabrication, and integration with complementary technologies. With continued advancements, these sensors could play a pivotal role in addressing environmental challenges.

## Figures and Tables

**Figure 1 sensors-24-06150-f001:**
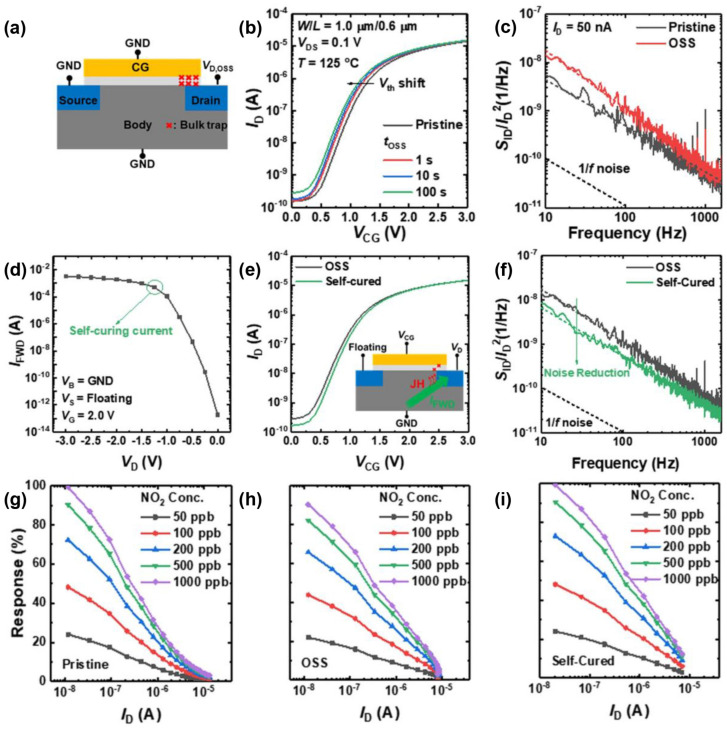
Recovery assisted by a self–curing method. (**a**) Diagram of gate oxide degradation due to OSS. (**b**) I_D_–V_CG_ versus t_OSS_. (**c**) S_ID_/I_D_^2^ before and after OSS. (**d**) V_D_–I_FWD_. (**e**) I_D_–V_CG_ of OSS and after a self–curing method. (**f**) S_ID_/I_D_^2^ of OSS and after self–curing. Response characteristics of the (**g**) pristine, (**h**) OSS–damaged, and (**i**) self–cured sensors. Reproduced with permission [51]. Copyright 2022, Springer.

**Figure 2 sensors-24-06150-f002:**
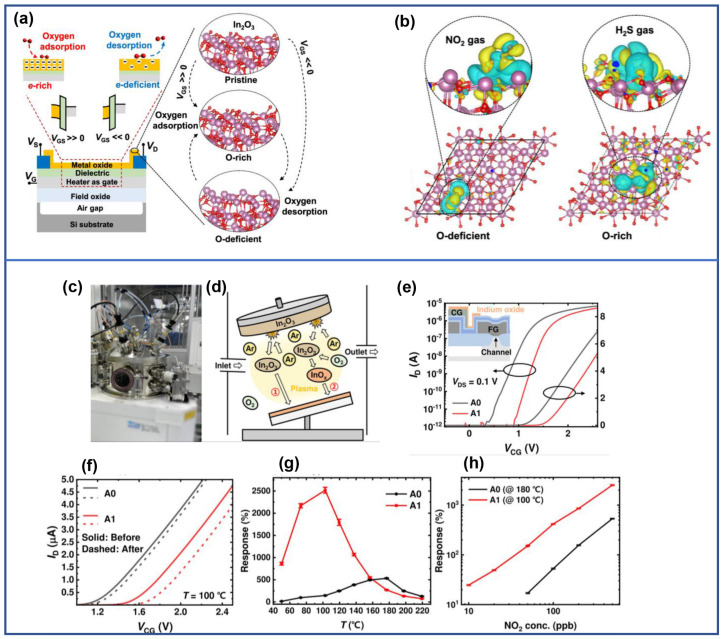
(top graph side) Reconfigurable control of oxygen content. (**a**) Schematic diagrams illustrating the band structure and electron concentration. (**b**) Adsorption model of NO_2_ and H_2_S. Reproduced with permission [52]. Copyright 2023, American Chemical Society. (bottom graph side) Hydroxy–rich–surface In_2_O_3_ gas sensor. (**c**) Photo of the magnetron RF sputtering. (**d**) Schematic representation of the sputtering process. (**e**) Transfer curves of two FET–type gas sensors. (**f**) Transfer characteristics corresponding to the response to NO_2_. (**g**) The response to 500 ppb NO_2_ versus operating temperature. (**h**) The response versus NO_2_ concentration. Reproduced with permission [53]. Copyright 2023, American Chemical Society.

**Figure 3 sensors-24-06150-f003:**
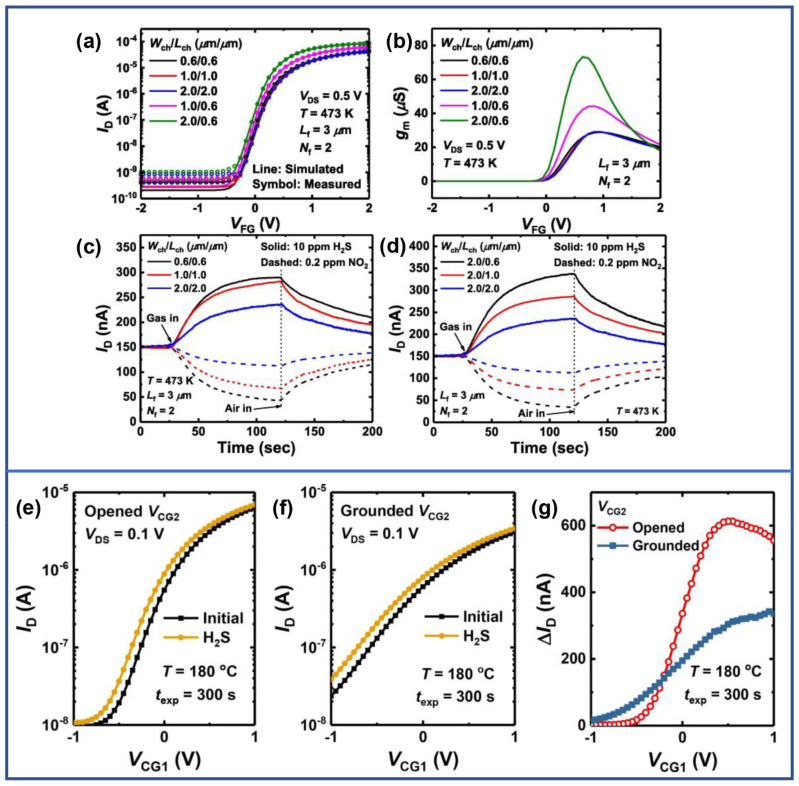
(top graph side) Si–based FET gas sensor. (**a**) I_D_–V_G_ curves and (**b**) gm–V_G_ curves. (**c**) Transient responses to NO_2_ and H_2_S at fixed W_ch_/L_ch_ ratio. (**d**) Transient responses to NO_2_ and H_2_S at varying W_ch_/L_ch_ ratio. Reproduced with permission [54]. Copyright 2023, Elsevier. (bottom graph side) H_2_S gas sensor based on polysilicon control–gate FET–type gas sensor. I_D_–V_CG1_ curves with (**e**) opened, and (**f**) grounded V_CG2_, respectively. (**g**) ΔI_D_ versus V_CG1_. Reproduced with permission [55]. Copyright 2023, Elsevier.

**Figure 4 sensors-24-06150-f004:**
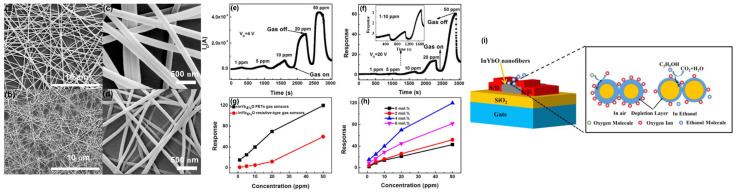
FET and resistive gas sensor based on Yb–doped In_2_O_3_. SEM images of the (**a**,**c**) as–electrospun, and (**b**,**d**) annealed 4% Yb–doped In_2_O_3_ nanofibers. Dynamic response of the (**e**) FET, and (**f**) resistive sensors. (**g**) Response comparison between the FET and resistive sensors. (**h**) Response versus Yb doping concentration. (**i**) Schematic diagram of sensing mechanism. Reproduced with permission [59]. Copyright 2020, American Chemical Society.

**Figure 5 sensors-24-06150-f005:**
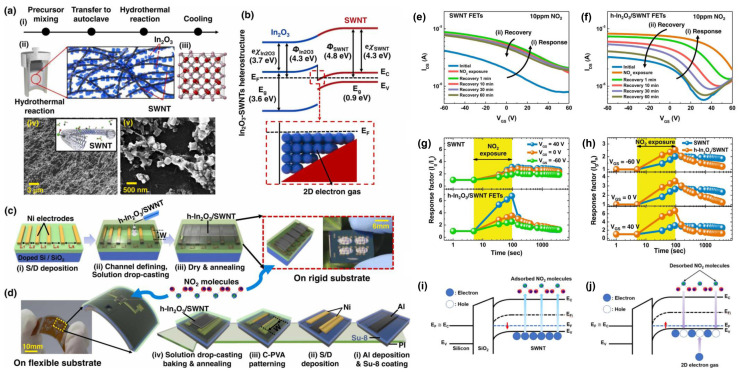
Gas sensing properties of the h–In_2_O_3_/SWNT TFT. (**a**) Preparation process of h–In_2_O_3_/SWNT and corresponding SEM images. (i) Preparation process of h–In_2_O_3_/SWNT. (ii) Schematic diagram of prepared h–In_2_O_3_/SWNT. (iii) The cubic crystal structure of h–In_2_O_3_/SWNT. SEM images of (iv) pure SWNTs and (v) h–In_2_O_3_/SWNT. (**b**) Band structure. Fabrication of gas sensors on the rigid (**c**) and flexible (**d**) substrates. Transfer characteristics of (**e**) SWNT FETs and (**f**) h–In_2_O_3_/SWNT FETs with response to NO_2_. (**g**) Response values obtained at dynamic measurements. (**h**) Comparison of response values to different V_GS_. Band structures’ diagrams of NO_2_ (**i**) response and (**j**) recovery. Reproduced with permission [62]. Copyright 2022, Elsevier.

**Figure 6 sensors-24-06150-f006:**
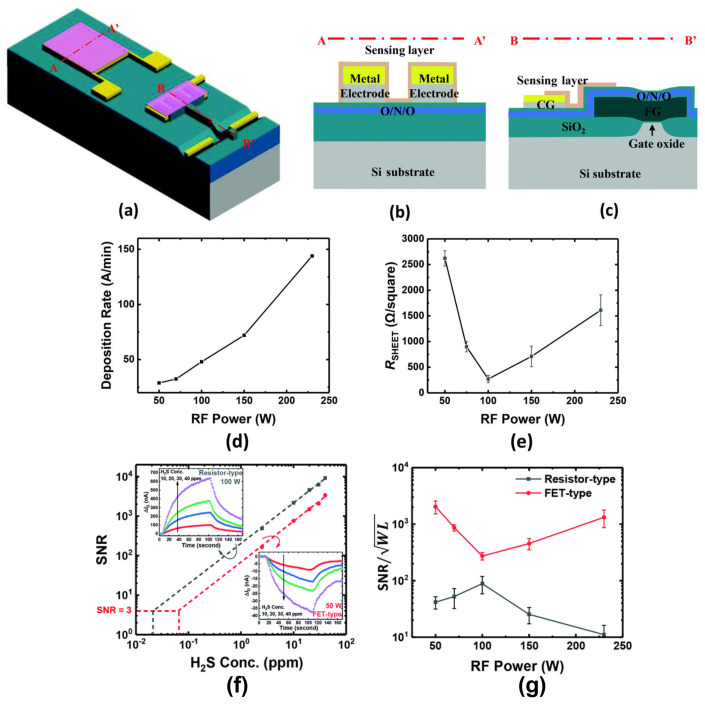
Influence of the deposition rate on signal–to–noise ratio. (**a**–**c**) Schematic diagram of resistive and FET sensors. (**d**) Deposition rate versus the RF sputtering power. (**e**) Resistivity versus the RF sputtering power. (**f**) The signal–to–noise ratio versus H_2_S concentration. (**g**) The signal–to–noise ratio per unit channel area versus the RF sputtering power. Reproduced with permission [47]. Copyright 2020, Royal Society of Chemistry.

**Figure 7 sensors-24-06150-f007:**
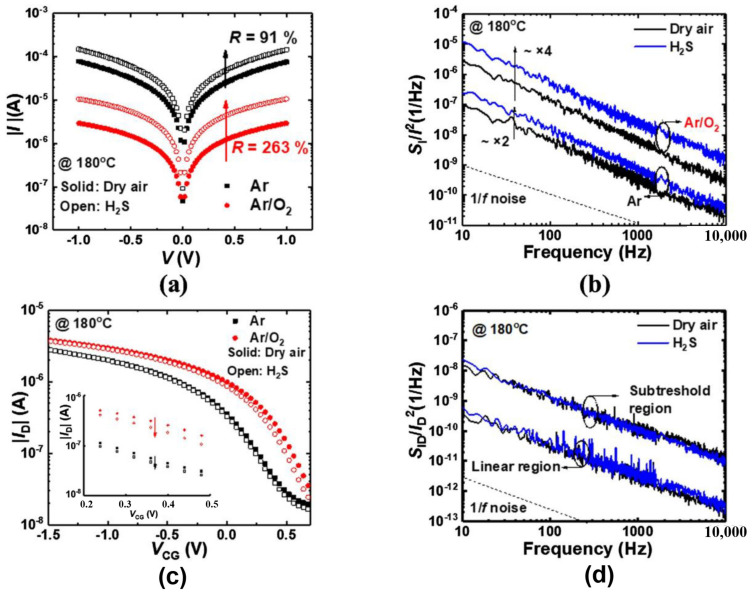
Characteristics of low–frequency noise for In_2_O_3_–based gas sensors. (**a**) Transfer characteristics of the resistive sensors in response to 50 ppm H_2_S. (**b**) The normalized low–frequency noise spectra of the resistive sensors with response to 50 ppm H_2_S. (**c**) Transfer characteristics of the FET sensors in response to 50 ppm H_2_S. (**d**) The normalized low–frequency noise spectra of the FET sensors with response to 50 ppm H_2_S. Reproduced with permission [48]. Copyright 2020, Elsevier.

**Figure 8 sensors-24-06150-f008:**
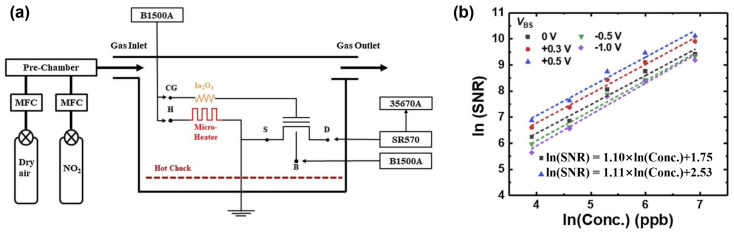
(**a**) Schematic of gas sensing and low–frequency noise measurement system. (**b**) Signal–to–noise ratio versus NO_2_ concentrations at logarithmic scale. Reproduced with permission [49]. Copyright 2021, Elsevier.

**Figure 9 sensors-24-06150-f009:**
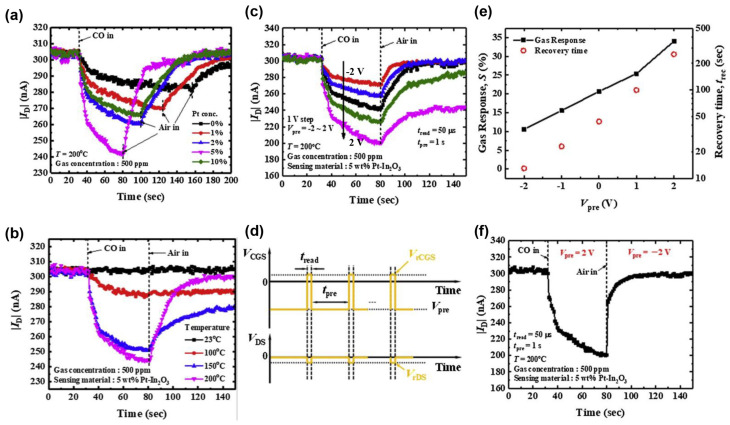
Si MOSFET gas sensor based on Pt–In_2_O_3_ with reduced recovery time. (**a**) The manufactured MOSFET’s transient CO response versus Pt concentration. (**b**) The transient CO response of the 5 wt% Pt–In_2_O_3_ sensor versus temperature. (**c**) The transient CO response of the 5 wt% Pt–In_2_O_3_ sensor versus V_pre_. (**d**) Pre–bias measuring pulse scheme. (**e**) Response values and recovery times versus V_pre_. (**f**) Enhanced response and recovery process by V_pre_. Reproduced with permission [46]. Copyright 2019, Elsevier.

**Figure 10 sensors-24-06150-f010:**
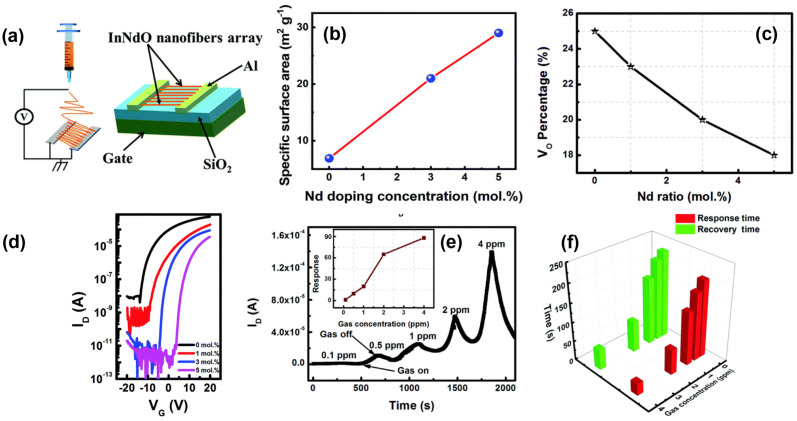
The Nd–doped In_2_O_3_ FET–based acetone sensor. (**a**) The electrospinning schematic diagram. (**b**) The correlation between specific surface area of InNdO nanofibers and Nd doping concentration. (**c**) Oxygen vacancy concentrations for varying Nd contents. (**d**) Transfer characteristics versus Nd doping concentrations. (**e**) The InNd_3%_O nanofiber FETs’ dynamic response. (**f**) Response and recovery times versus acetone concentrations. Reproduced with permission [61]. Copyright 2022, Royal Society of Chemistry.

**Figure 11 sensors-24-06150-f011:**
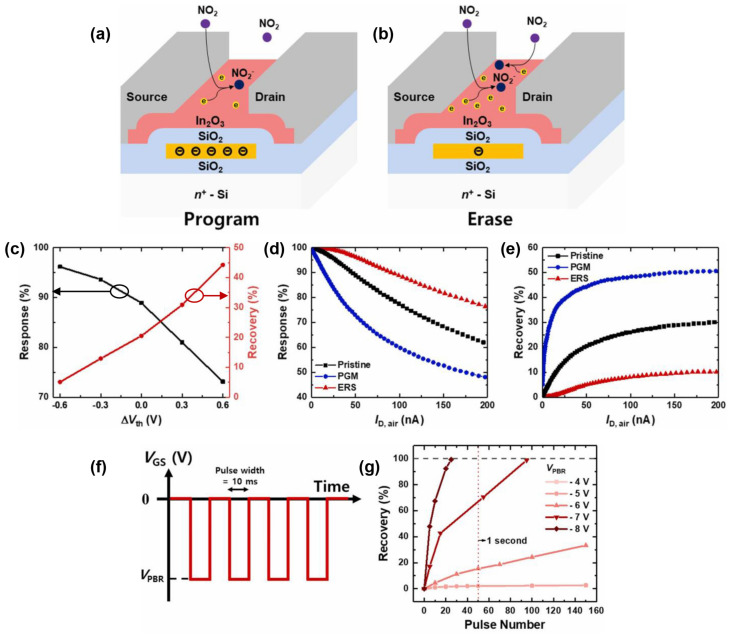
Fast response/recovery NO_2_ sensor based on In_2_O_3_ TFT. Diagram of the NO_2_ gas detecting system in the (**a**) program and (**b**) erase states. (**c**) Response and recovery behaviors versus ΔV_th_. (**d**) Response and (**e**) recovery behaviors versus I_D_. (**f**) The PBR method’s pulse scheme. (**g**) Recovery behaviors versus V_PBR_. Reproduced with permission [63]. Copyright 2023, Elsevier.

**Figure 12 sensors-24-06150-f012:**
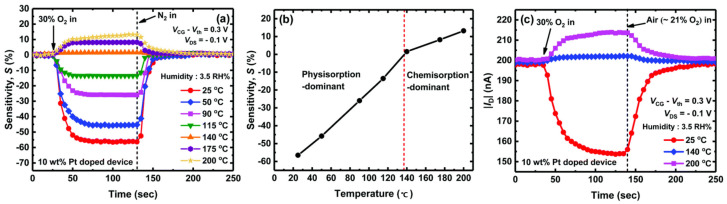
Physisorption in an oxygen sensor of the high–performance In_2_O_3_ FET type operating at room temperature. (**a**) Dynamic response of 30% O_2_. (**b**) Sensitivities versus temperature. (**c**) Dynamic response of 30% O_2_ with ∼21% O_2_ set as the reference gas. Reproduced with permission [45]. Copyright 2018, Royal Society of Chemistry.

**Figure 13 sensors-24-06150-f013:**
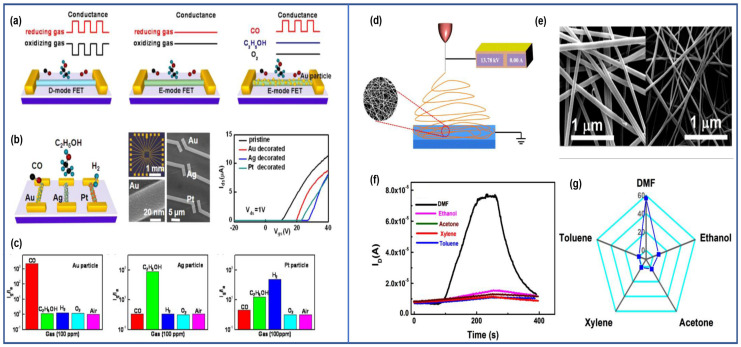
(left graph side) (**a**) Diagram illustrating the sensor configuration for the “one lock to one key” design concept. (**b**) Hybrid Mg–doped In_2_O_3_ NW FETs decorated with Au, Ag, and Pt nanoparticles, respectively. (**c**) The sensitivity of three FETs decorated with Au, Ag, and Pt nanoparticles in response to different gases. Reproduced with permission [74]. Copyright 2013, American Chemical Society. (right graph side) (**d**) Preparation of electrospinning. (**e**) SEM images of Yb–doped nanofibers before and after the annealed process. (**f**) Transient response of the Yb–doped In_2_O_3_ TFT towards different gases. (**g**) The selectivity of the sensor. Reproduced with permission [60]. Copyright 2020, Elsevier.

**Figure 14 sensors-24-06150-f014:**
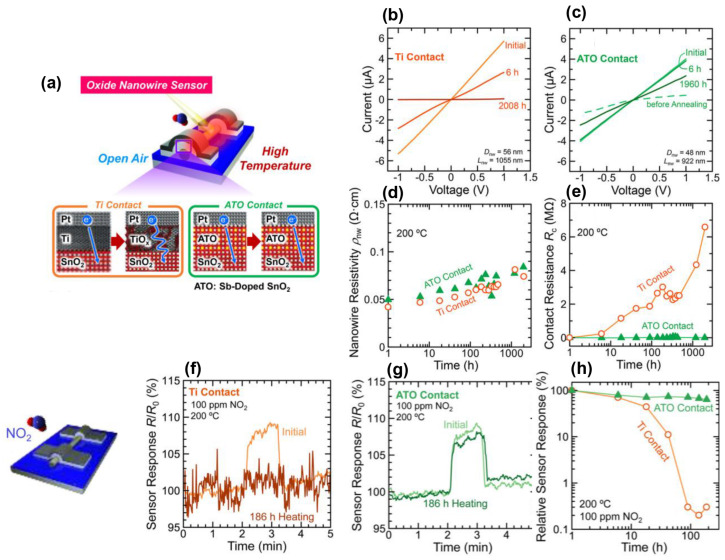
Long–term stability improved by replacing traditional metal contacts with heavily doped metal oxides. (**a**) Diagram of reasons for improved long–term stability. I–V curves of (**b**) Ti and (**c**) ATO contact devices. (**d**) The resistivity of metal oxide nanowires versus the aging time. (**e**) The contact resistance of metal oxide nanowires versus the aging time. The sensor response towards NO_2_ versus after 186 h aging of (**f**) Ti and (**g**) ATO contact devices. (**h**) The varying response values that change over aging time. Reproduced with permission [75]. Copyright 2017, American Chemical Society.

**Figure 15 sensors-24-06150-f015:**
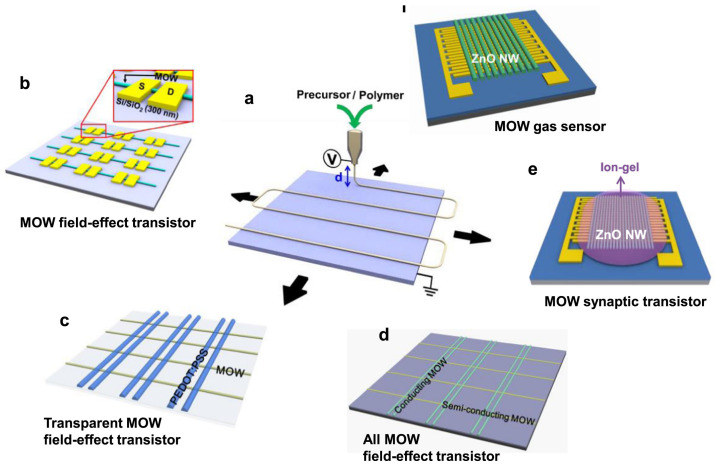
Diagram of the nanoscale MOW circuits printed directly on substrate. (**a**) Printing under digital control. (**b**) Metal oxide nanowire FET. (**c**) Transparent metal oxide nanowire FET. (**d**) All metal oxide nanowire FET. (**e**) Metal oxide nanowire synaptic transistors. (**f**) Metal oxide nanowire gas sensors. Reproduced with permission [76]. Copyright 2019, Elsevier.

**Figure 16 sensors-24-06150-f016:**
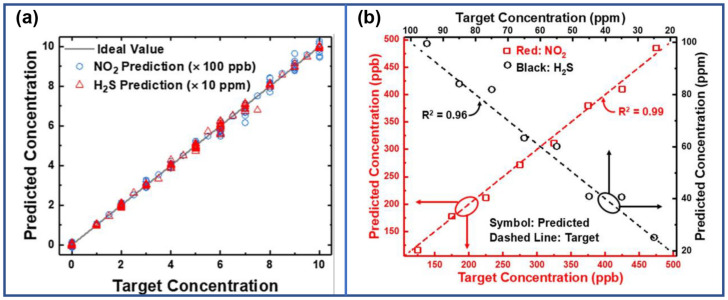
(left graph side) (**a**) Predicted concentration for NO_2_ and H_2_S utilizing the RNN train over 500 epochs compared to the target concentration. Reproduced with permission [77]. Copyright 2021, Elsevier. (right graph side) (**b**) Predicted H_2_S and NO_2_ concentrations through SNNs. Reproduced with permission [78]. Copyright 2021, Elsevier.

**Table 1 sensors-24-06150-t001:** In_2_O_3_ transistors with high mobility.

Type	Preparation Method	Mobility/μ cm^2^v^−1^s^−1^	Refs.
Commercial IGZO thin film	PVD	10	/
Unpassivated In_2_O_3_ thin film	PVD	85	[81]
In_2_O_3_ nanowires	CVD	>200	[82]
Zn–doped In_2_O_3_ nanowires	CVD	139	[83]
PVA–doped In_2_O_3_ thin film	Solution processed	4	[84]
PEI–doped In_2_O_3_ thin film	Solution processed	9	[84]

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
