# Peer review of "Gas Sensing Properties of Indium–Oxide–Based Field–Effect Transistor: A Review"

_sensors, 2024, doi:10.3390/s24186150_

Round 1

Reviewer 1 Report

Comments and Suggestions for Authors

This review provides a brief overview of the latest advances in indium oxide-based field-effect transistors as gas sensors, covering the sensor's advantages, types of transducers, methods for optimizing sensing performance, sensing mechanisms, and potential practical applications. It also discusses the challenges faced by indium oxide transistor gas sensors and offers insights for future research. However, I have some questions that might be helpful for researchers in the field:

1. In the field of metal oxide semiconductor gas sensors, materials such as tin oxide, zinc oxide, and copper oxide are more extensively studied, and research on them as field-effect transistors is also considerable. Is it necessary to specify the advantages of indium oxide more clearly?

2. Resistive sensors are the most widely used transducers in gas sensors and have been commercially developed on a large scale. Can field-effect transducer sensors change this status in the future?

3. Can the advantages of indium oxide field-effect transistors be more intuitively demonstrated by comparing performance parameters in a tabular format?

Comments on the Quality of English Language

The quality of English Langage is quite good.

Author Response

Comments to the Author: This review provides a brief overview of the latest advances in indium oxide-based field-effect transistors as gas sensors, covering the sensor's advantages, types of transducers, methods for optimizing sensing performance, sensing mechanisms, and potential practical applications. It also discusses the challenges faced by indium oxide transistor gas sensors and offers insights for future research. However, I have some questions that might be helpful for researchers in the field.

Response:

Thank you for your positive comments very much. All the authors tried their best to improve the quality of this manuscript according to your suggestions and comments.

  1. In the field of metal oxide semiconductor gas sensors, materials such as tin oxide, zinc oxide, and copper oxide are more extensively studied, and research on them as field-effect transistors is also considerable. Is it necessary to specify the advantages of indium oxide more clearly?

Reply 1:

Thank you for your suggestion. In the field of gas sensing, indium oxide has several notable advantages over tin oxide, zinc oxide, and copper oxide, including high electrical conductivity and sensitivity, a wide range of gas response, good stability and durability, lower operating temperature, and high surface reactivity. In the revised content, we emphasize the high conductivity of indium oxide due to its high carrier concentration and mobility.

The description of the modified location and content.

Line 65-68. In2O3 is a transparent metal oxide semiconductor with a forbidden band width of 3.4-3.75 eV and retains higher carrier concentration and carrier mobility compared to other oxide semiconductors.was replaced by “Indium oxide (In₂O₃) is a transparent metal oxide semiconductor with a bandgap of 3.4 to 3.75 eV. It surpasses other common oxide semiconductors, such as SnO₂, ZnO, and CuO, in carrier concentration and mobility. This high electrical conductivity re-sults in exceptional gas sensitivity, particularly in low concentrations (ppm to ppb).”

  1. Resistive sensors are the most widely used transducers in gas sensors and have been commercially developed on a large scale. Can field-effect transducer sensors change this status in the future?

Reply 2:

In display technology, the most advanced industrial IGZO (Indium Gallium Zinc Oxide) is based on indium oxide materials. Therefore, a mature industrial system is beneficial for shortening the research time for indium oxide-based materials. Therefore, summarizing advancements in indium-based oxide gas sensors can facilitate broader deployment of In₂O₃ FET gas sensor.

The description of the modified location and content.

Line 75-78. “The optical transparency of In₂O₃ makes it ideal for transparent conductive layers, with indium-based oxides like IGZO being advanced materials in display technologies. Summarizing advancements in indium-based oxide gas sensors can facilitate broader deployment of In₂O₃ FET gas sensors.” was added.

  1. Can the advantages of indium oxide field-effect transistors be more intuitively demonstrated by comparing performance parameters in a tabular format?

Reply 3:

According to the recommendations, we listed the mobility of three types of transistors based on the greatest advantage of indium oxide over other metal oxides, its high carrier concentration and mobility, which is beneficial for gas sensing.

line 734-735. “Table 1 here lists several types of In2O3 transistors with high mobility.” was added.
line 740.
“Table 1” was added.

line 999-1013. reference [82-87] was added.

82      Zhao, H.; Li, J.; She, X.; Chen, Y.; Wang, M.; Wang, Y.; Du, A.; Tang, C.; Zou, C.; Zhou, Y. Oxygen Vacancy-Rich Bimetallic Au@Pt Core-Shell Nanosphere-Functionalized Electrospun ZnFe2O4 Nanofibers for Chemiresistive Breath Acetone Detection. ACS Sens. 2024, 9, 2183-2193.

83      Yu, C.; Liu.; J. Zhao, H.; Wang, M.; Li, J.; She, X.; Chen, Y.; Wang, Y.; Liu, B.; Zou, C.; He, Y.; Zhou, Y. Sensitive Breath Acetone Detection Based on α-Fe2O3 Nanoparticles Modified WO3 Nanoplate Heterojunctions. IEEE Transactions on Instrumentation and Measurement. 2024.

84      Ghediya, P. R.; Magari, Y.; Sadahira, H.; Endo, T.; Furuta, M.; Zhang, Y.; Matsuo, Y.; Ohta, H. Reliable Operation in High-Mobility Indium Oxide Thin Film Transistors. Small Methods. 2024, 2400578.

85      Shen, G.; Liang, B.; Wang, X.; Chen, P. C.; Zhou, C. Indium oxide nanospirals made of kinked nanowires. Acs Nano. 2011, 5, 2155-2161.

86      Singh, N.; Yan, C.; Lee, P. S. Room temperature CO gas sensing using Zn-doped In2O3 single nanowire field effect transistors. Sensors and Actuators B: Chemical. 2010, 150, 19-24.

87     Huang, W.; Zeng, L.; Yu, X.; Guo, P.; Wang, B.; Ma, Q.; Chang, R. P. H.; Yu, J.; Bedzyk, M. J.; Marks, T.J.; Facchetti, A. Metal Oxide Transistors via Polyethylenimine Doping of the Channel Layer: Interplay of Doping, Microstructure, and Charge Transport. Advanced Functional Materials. 2016, 26, 6179-6187.

Reviewer 2 Report

Comments and Suggestions for Authors

The manuscript "Gas Sensing Properties of Indium-Oxide-Based Field Effect Transistor: A Review" provides an overview of the latest advancements in indium oxide (In₂O₃) FETs for gas sensing, in terms of sensing performance parameters, device structures, and how to optimize in the future. In general, the review is complete, the organization of the review is decent, and the review itself is thorough. However, there are critical issues that need to be addressed, in terms of content and language quality before it can be reconsidered for publication.

1. Structure of the Review: In Part 3, the authors describe indium oxide devices themselves, their different structures and their gas sensing mechanisms, which seems more appropriate as introductory content. Meanwhile, Part 2 covers key gas sensing parameters. To improve the logical flow, I recommend reorganizing the review by moving Part 3 before Part 2. This would allow readers to first understand the basic device structures and mechanisms before diving into performance metrics.

2. Incomplete Explanation of Gas Sensing Mechanisms: In part 3, line 454-468, the explanation of gas sensing mechanism between resistive device and FET is incomplete and does not fully capture the differences between the two. Firstly, the working principal of both resistor sensor and FET sensor (TFT version, sensing layer is also FET channel) are both relying on changing of the resistance of the sensitive layer (conducting channel), which makes the gas signal. The gas itself interacts with the surface of the sensitive layer in terms of charge transfer (TFT version), screening (or field-effect, floating gate version). Physical adsorption or chemical reaction are involved in both resistor sensors and FET sensors.

The main differences in my opinion comes from the different numbers of terminals between these 2 devices, i.e. 2 terminals resistor vs 3 terminal FETs. In FETs, one can always adjust Gate voltage to tune the optimal device condition to achieve best sensitivity. Together with other differences, as mentioned in the papers, such as multi-parameter indicators, mark clear advantage of FET sensor vs resistor sensor. This section should be revised to clearly explain these differences and emphasize the advantages of FET sensors over resistive sensors.

3. Comparison of FET Gas Sensors: The review lacks a direct comparison of the two main types of FET gas sensors: those based on the FET channel and those utilizing the FET floating gate. The authors mention various studies but do not provide their own analysis of the pros and cons of these types. Including a detailed comparison would make the review more insightful and valuable for readers.

4. Citations in Future Perspectives: In Part 5, which discusses challenges and future perspectives, the authors present various opinions and predictions about the development of indium oxide FET gas sensors. If these viewpoints are derived from existing literature, appropriate citations should be included to credit the original sources. This will help differentiate the authors' original contributions from previously published ideas, which is crucial for a insightful review paper.

5. Language and Writing Quality: The manuscript has numerous language issues, particularly in the introduction, conclusion, and abstract sections. While this review highlights some of these issues, I strongly recommend that the authors thoroughly revise the language and improve the writing quality throughout the paper. A professional language edit would greatly enhance the clarity and impact of the review.

Comments on the Quality of English Language

1. Long sentences need to revised. In introduction part, for instance, line 54-57, line 60-63, these are very long sentences. Please consider trim them and enhance the clarity.

2. Line 64, "forbidden band width" ->"bandgap"

3. Some words, such as "ability to move easily", do you mean "portable"? 

4. “being integrable”, how about "integration compatible"?

There are really a lot of language issues in this paper, please revise and improve the writing quality throughout the paper.

Author Response

Reviewer #2:

Comments to the Author: The manuscript "Gas Sensing Properties of Indium-Oxide-Based Field Effect Transistor: A Review" provides an overview of the latest advancements in indium oxide (In₂O₃) FETs for gas sensing, in terms of sensing performance parameters, device structures, and how to optimize in the future. In general, the review is complete, the organization of the review is decent, and the review itself is thorough. However, there are critical issues that need to be addressed, in terms of content and language quality before it can be reconsidered for publication.

Response:

After carefully reviewing these comments, we believe that the flaws you pointed out have been very helpful to our research. We have had thorough discussions and made corresponding revisions to the article based on your feedback.

  1. Structure of the Review: In Part 3, the authors describe indium oxide devices themselves, their different structures and their gas sensing mechanisms, which seems more appropriate as introductory content. Meanwhile, Part 2 covers key gas sensing parameters. To improve the logical flow, I recommend reorganizing the review by moving Part 3 before Part 2. This would allow readers to first understand the basic device structures and mechanisms before diving into performance metrics.

Reply 1:

Thank you very much for your useful suggestions. We have swapped the content of sections 2 and 3. The article now introduces the fundamental transducer structure and mechanism first, followed by a discussion of the key gas sensing parameters. The content of the first three chapters of the paper has been reorganized.

The description of the modified location and content.

Line 85-609. The content has been reorganized according to the suggestion.

  1. Incomplete Explanation of Gas Sensing Mechanisms: In part 3, line 454-468, the explanation of gas sensing mechanism between resistive device and FET is incomplete and does not fully capture the differences between the two. Firstly, the working principal of both resistor sensor and FET sensor (TFT version, sensing layer is also FET channel) are both relying on changing of the resistance of the sensitive layer (conducting channel), which makes the gas signal. The gas itself interacts with the surface of the sensitive layer in terms of charge transfer (TFT version), screening (or field-effect, floating gate version). Physical adsorption or chemical reaction are involved in both resistor sensors and FET sensors.

The main differences in my opinion comes from the different numbers of terminals between these 2 devices, i.e. 2 terminals resistor vs 3 terminal FETs. In FETs, one can always adjust Gate voltage to tune the optimal device condition to achieve best sensitivity. Together with other differences, as mentioned in the papers, such as multi-parameter indicators, mark clear advantage of FET sensor vs resistor sensor. This section should be revised to clearly explain these differences and emphasize the advantages of FET sensors over resistive sensors

Reply 2:

Thank you very much for pointing out where our explanation was unclear. After further consideration, we have corrected this part of the mechanism and hope it is now clearer and more accurate.

The description of the modified location and content.

Line 454-468. The paragraph is entirely replaced by “Some differences exist in the operational mechanisms of In2O3 resistive and FET gas sensors, although both rely on the interaction between the thin film sensing layers and target gases. For resistive gas sensors, the gas molecules react with the metal ox-ides, altering the resistance and thereby detecting the gas concentration. This change is measured by tracking the variation in resistance values. In contrast, for FET gas sen-sors, the performance changes in the gas-affected thin film sensing layer are mani-fested in charge transfer of TFT devices and screening of FET and floating-gate based devices under the influence of an electric field. These changes are measured by dy-namic changes in current or transient changes in voltage. Compared to the wide ap-plication scenarios of resistive sensors, FET sensors are more suitable for high sensitiv-ity and high selectivity gas detection. Additionally, they generally have lower power consumption and better stability. Here, we elaborate on the progress of In2O3 FET gas sensors based on two mainstream transducers (MOSFET and TFT).” and the paragraph is located on line 109-121.

  1. Does the test temperature affect the sensing performance? Because this is very important to the actual application of the sensor. 3. Comparison of FET Gas Sensors: The review lacks a direct comparison of the two main types of FET gas sensors: those based on the FET channel and those utilizing the FET floating gate. The authors mention various studies but do not provide their own analysis of the pros and cons of these types. Including a detailed comparison would make the review more insightful and valuable for readers.

Reply 3:

Thank you for your suggestion. According to the recommendations, we compared floating-gate FET devices with more traditional FET channel-based devices. We explained the working mechanism of current conduction affected by the floating gate. Additionally, we discussed the manufacturing complexity, scalability, gas sensing response, signal-to-noise ratio, and detection limits of the two types of sensors. We hope this information is more valuable to the readers.

The description of the modified location and content.

Line 295-309. “Those sensors based on FET channels typically adopt a traditional structure consisting mainly of a source, drain, and gate, and the conductivity of the FET channel is controlled by the gate voltage. In contrast, an additional insulated floating gate is add-ed on top of the traditional structure for FET devices based on floating-gate, allowing charges to accumulate or release on the floating gate, which directly affects the current conduction. Therefore, the carrier transport in the channel is regulated by both control voltage, VCG and floating voltage, VFG. Correspondingly, floating-gate FET gas sensors are more suitable for high-sensitivity gas detection. On the other hand, TFTs as the most common representative utilized in the fielding of gas sensing, acquire a simpler structure, while floating-gate FET sensors typically employ MOSFETs, which involve more complex manufacturing processes. MOSFETs are regularly smaller than TFTs, making them more suitable for integration. However, TFTs exhibit larger responses and higher signal-to-noise ratios, resulting in lower detection limits for gas sensing. Consequently, the specific selection between MOSFETs and TFTs is determined by considering the requirements of the actual application.” was added.

  1. 4. Citations in Future Perspectives: In Part 5, which discusses challenges and future perspectives, the authors present various opinions and predictions about the development of indium oxide FET gas sensors. If these viewpoints are derived from existing literature, appropriate citations should be included to credit the original sources. This will help differentiate the authors' original contributions from previously published ideas, which is crucial for a insightful review paper.

Reply 4:

Thank you very much for your friendly and valuable suggestions. We have cited the sources for the viewpoints derived from existing literature.

Line 737 [60, 63, 77] was updated.

Line 745 [60-62] was updated.

Line 754 [46] was updated.

Line 756 [47, 51, 53] was updated.

Line 761 [78-81] was updated.

Line 766 [63] was updated.

Line 772 [76] was updated.

Line 779 [54, 77] was updated.

Line 786 [47, 62, 64] was updated.

Line 794 [57, 75, 82-83] was updated.

5 Language and Writing Quality: The manuscript has numerous language issues, particularly in the introduction, conclusion, and abstract sections. While this review highlights some of these issues, I strongly recommend that the authors thoroughly revise the language and improve the writing quality throughout the paper. A professional language edit would greatly enhance the clarity and impact of the review.

Reply 5:

Thank you for your suggestions. We have had a native English speaker make extensive revisions to the text, including but not limited to the issues you raised. We have improved the English in the scientific writing according to the requirements. Enclosed please find a clear copy with changes marked by highlights for your reference.

The description of the modified location and content.

Line 54-57 “The multi-parameters of FETs, such as transient threshold voltage (Vth), sub-threshold swing (SS), switching ratio (Ion/Ioff), carrier concentration (μ), transconductance (gm), saturated output current, etc., can be extracted and beneficial to the distinction and identification of mixed gases.” was replaced by “The extraction multiple parameters of FETs, including transient threshold voltage (Vth), sub-threshold swing (SS), switching ratio (Ion/Ioff), carrier concentration (μ), transconductance (gm), saturation output current, etc., can be advantageous for the identification of mixed gases.”

Line 60-63 “Moreover, a back-gate top-contact structure is commonly adopted for FET gas sensors. The reason why the top-contact structure is not selected is that gas-sensitive materials routinely need to be exposed to the gas environment and the direct contact of gas-sensitive materials and target gases can manage faster measurement.” was replaced by “Moreover, a back-gate top-contact configuration is commonly adopted for FET gas sensors. Since gas-sensitive materials require frequent exposure to the gas environ-ment, the top-contact structure is less selected. The direct interaction between gas-sensitive materials and target gases facilitates expedited measurement.”

Line 64 “forbidden band width”->bandgap”

Line 35 “ability to move easily”->portable

Line 45 “being integrable” -> “integration compatible”

Round 2

Reviewer 2 Report

Comments and Suggestions for Authors

In general, the authors addressed the issues I mentioned last time, the review paper now looks better than last time. I recommend its publication this time.

Comments on the Quality of English Language

There are still some very long sentences, consider trimming them into separate sentences. 

Author Response

In general, the authors addressed the issues I mentioned last time, the review paper now looks better than last time. I recommend its publication this time.

Response:

Thank you very much for your constructive comments. We have done our utmost to improve the quality of our manuscript.

There are still some very long sentences, consider trimming them into separate sentences.

Reply:

we have revised long sentences that may have been unclear or potentially ambiguous, aiming for more concise and accurate expressions. Additionally, we have replaced some words to enhance the logical flow and precision of our writing. Enclosed please find a clear copy with changes marked by highlights for your reference.

The description of the modified location and content.

Line 97-99 “Moreover, In2O3 nanocomposites……” was replaced by “Furthermore, In2O3 nanocomposites hold significant promise for improving gas sensor performance, particularly when combined with precious metals and metal oxides.”

Line 133-137 “In 2020, Wonjun Shin et al. explored……” was replaced by “In 2020, Wonjun Shin et al. investigated the impact of deposition conditions on the signal-to-noise ratio of gas sensors. The MOSFET gas sensors were based on In2O3 thin films of the same thickness (30 nm) prepared by magnetron sputtering at different radio frequency powers [48]. Within the low power range (50-100 W), increased power results in larger In2O3 grains with fewer grain boundaries, consequently reducing film resistivity.”

Line 140-145 “Nevertheless, variations in RF sputtering power do not affect……” was replaced by “Nevertheless, variations in RF sputtering power do not affect the low-frequency noise (LFN) characteristics in MOSFET gas sensors, as the channel of the MOSFET deter-mines the low-frequency noise characteristics. The LFN characteristics of MOSFET-based gas sensors depend on factors such as the channel, operating region (linear region/subthreshold region/saturation region), and biases between the source and drain.”

Line 149-151 “Therefore, the highest signal-to-noise ratio……” was replaced by “Therefore, the highest signal-to-noise ratio is attained at the power level where maxi-mum gas response occurs, attributed to the increased film defects that facilitate gas adsorption.”

Line 157-160 “The MOSFET based on the thin film prepared in the Ar/ O2……” was replaced by “The MOSFET with the thin film prepared in an Ar/O2 mixed atmosphere demonstrated a stronger response to H2S without an increase in additional noise, compared to the MOSFET with the film prepared in pure Ar.”

Line 162-165 “In the following year, their group continued the investigation……” was replaced by “In the following year, their group continued the investigation the influence of the body-source junction bias voltage on NO2 MOSFET sensors utilizing a 12 nm In2O3 thin film as the sensitive material. The low-frequency noise characteristics were found to be affected by the polarity of the body-source junction bias (VBS).”

Line 183-185 “The issues stemming from threshold voltage shift……” was replaced by “The issues stemming from threshold voltage shift, increased noise, and reduced response to NO2 were induced by off-state stress- related damage. They could all be rectified using this approach, as illustrated in Figure 1d-i [52].”

Line 234-236 “Jinwoo Park et al. replaced the previously commonly adopted metal gate……” was replaced by “Jinwoo Park et al. replaced the commonly adopted metal gate in MOSFET gas sensors constructed with a 15 nm thick In2O3 film with a polycrystalline silicon gate.”

Line 254-262 “The TFT based on this porous In2O3 film achieved selective detection……” was replaced by “Although the porous film structure resulted in suboptimal TFT electrical performance. The TFT based on this porous In2O3 film achieved selective detection of ethanol at room temperature [57]. The influence of different annealing times at 400°C on the electrical performance of TFTs based on spin-coated oil soluble In2O3 nanoparticles, was report-ed by Wang et al. in 2015. The In2O3 TFT annealed for 10 minutes exhibited the highest output current in air [58]. In 2017, Shariati Mohsen demonstrated that the morphology of Sn-doped In2O3 nanowires was affected by annealing temperature, which in turn influenced the electrical performance of the TFT as well as the recovery process following H2S exposure.”

Line 319-322 “Larger responsivity, normally originated from abundant adsorption sites……” was replaced by “Increased responsivity, typically arising from abundant adsorption sites, is frequently observed in research; however, these sites may also indicate a greater presence of defects in the FET channel, potentially resulting in increased noise and degradation of electrical performance.”

Line 399-401 “The highest SNR of the In2O3 thin film……” was replaced by “The highest SNR observed in the In2O3 thin film-based resistive sensor occurred at the sputtering power that yielded the lowest resistivity, despite the device's response not being maximal under these conditions.”

Line 448-450 “The researches on enhancing signal-to-noise ratio……” was replaced by “The research conducted by the group to enhance the signal-to-noise ratio evolved from material deposition to transducer comparison, ultimately culminating in the optimization of the transducer's operating conditions.”

Line 458-460 “In 2019, Seongbin Hong et al. applied Pt-modified In2O3……” was replaced by “In 2019, Seongbin Hong et al. applied Pt-modified In2O3 to MOSFET utilizing inkjet printing technology. The optimal 5% Pt doping concentration and the best sensing temperature of 200°C for detecting CO were determined.”

Line 477-478 “The excellent electrical performance, including a high mobility……” was replaced by “The sensor exhibited outstanding electrical performance, characterized by a high mobility of 5.5 cm² V⁻¹s⁻¹ and an on/off current ratio of 10⁷.”

Line 518-520 “Porous indium oxide based TFT sensors achieved……” was replaced by “In 2012, Porous indium oxide based TFT sensors achieved acceptable ethanol selectivity at temperatures below 100°C. In 2013, gold nanoparticle-modified Mg-In2O3 nanowire FET arrays realized the recognition of CO in mixed gas at room temperature [57,75].”

Line 528-530 “In 2018, Seongbin Hong et al. prepared Pt-doped In2O3 MOSFET……” was replaced by “In 2018, Seongbin Hong et al. prepared Pt-doped In2O3 MOSFET O2 sensors utilizing inkjet printing. Oxygen sensing at room temperature was achieved after optimizing the Pt doping concentration.”

Line 533-536 “In 2020, an ethanol gas sensor based on……” was replaced by “In 2020, an ethanol gas sensor based on Yb-doped In2O3 (InYbO) nanofiber TFT achieved a detection limit of 1 ppm for ethanol at temperatures below 80°C. The 3% InNdO TFT managed a high response of 88 to 4 ppm acetone at room temperature, with a response time of 31 seconds and a recovery time of 53 seconds [60].”

Line 574-576 “The sensor exhibited a considerably higher response to ethanol……” was replaced by “The sensor demonstrated a significantly higher response to ethanol compared to ammonia and acetone, exhibiting minimal response variation within the relative humidity range of 40-60% [57].”

Line 596-599 “FET gas sensors can achieve improved selectivity not only……” was replaced by “FET gas sensors can achieve enhanced selectivity not only through material considerations but also through a synergistic combination of mechanism-based and material-based designs, contrasting with the extensively researched resistive sensors.”

Line 647-649 “Third, the properties are tunable such as conductivity and bandgap,……” was replaced by “Third, properties such as conductivity and bandgap are tunable and can be tailored by adjusting size, composition, and morphology, enabling optimization for specific application requirements.”

Line 656-659 “Traditional In2O3 FETs in D-mode respond to……” was replaced by “Traditional In2O3 FETs in D-mode respond to both oxidizing and reducing gases. However, the threshold voltage of the FET can be increased to form an E-mode by adjusting the Mg doping concentration, in which the device does not respond to oxidizing gases.”

Line 670-673 “This method involves high-pressure direct printing……” was replaced by “The method involves high-pressure direct printing of sacrificial polymer and precursor solutions onto a silicon substrate, followed by sintering to form the MOW array. The top contact electrode is fabricated using the conventional metal thermal evaporation process.”

Line 722-726 “In contrast to resistive gas sensors……” was replaced by “Unlike resistive gas sensors relying on transient dynamic response parameters such as response values and response times, FET gas sensors offer a wider range of extractable parameters from the transient transfer and output curves such as Saturation output current, switching ratio, threshold voltage, carrier concentration and mobility, sub-threshold swing, etc.”

Line 742-743 “One of the primary challenges is achieving high selectivity……” was replaced by “A principal challenge lies in achieving high selectivity for specific target gases while effectively minimizing interference from other environmental gases.”

Line 772-773 “While advancements have been made in miniaturizing In2O3 FET gas sensors……” was replaced by “Despite advancements in miniaturizing In₂O₃ FET gas sensors, additional efforts are required to integrate them into compact and portable devices for real-world applications.”

Line 779-780 “Improving the response time and recovery time is essential……” was replaced by “Enhancing both response and recovery times is crucial for applications that de-mand rapid detection and monitoring of gas concentrations.”

Line 792-794 “Material engineering and the internal electric field generated by……” was replaced by “Advances in material engineering, combined with the internal electric fields generated by field-effect properties, present opportunities to achieve sensing at lower temperatures.”
